# Position: Time-Series Foundation Models Require Explicit Domain-Level Benchmarks

**Md Asif Bin Syed** [* 1]  **Md Younus Ahamed** [* 2]  **Azmine Toushik Wasi** [3]

## Abstract

Time series foundation models (TSFMs) have demonstrated strong performance on established benchmarks such as GIFT-Eval, Monash, and TSFM-Bench. However, these benchmarks pool datasets from many domains with uneven representation, which can obscure performance within specific application areas such as healthcare, finance, nature, retail, and transport. The necessity for domain-specific evaluation arises from the inherent structural diversity of time series data: clinical records often feature irregular sampling and informative missingness; financial sequences are characterized by high noise and stochastic trajectories; and environmental data, such as energy and weather, are governed by deterministic physical laws and strong seasonal hierarchies. Motivated by this heterogeneity, **we argue that TSFMs require explicit domain-specific benchmarks** so practitioners can reliably assess a model's utility within their own application area. This is because cross-domain differences in data generation, sampling irregularity, and nonstationarity under concept drift fundamentally shape forecasting difficulty and failure modes. As a result, strong performance on aggregated leaderboards may not translate to reliable deployment within a specific domain. To test this, we evaluated seven TSFMs across 72 datasets from six domains and found substantial cross-domain variability. These findings confirm that global benchmark scores can be misleading and that domain-aware evaluations are essential for trustworthy TSFM selection.

[1]College of Computing, Gerogial Institute of Technology, Atlanta, GA 30332 [2]Lane Department of Computer Science and Electrical Engineering, West Virginia University, Morgantown, WV 26506, USA [3]Department of Industrial and Production Engineering, Shahjalal University of Science and Technology (SUST), Sylhet. Correspondence to: Md Asif Bin Syed <msyed70@gatech.edu>, Md Younus Ahamed <ma00087@mix.wvu.edu>.

*Proceedings of the $43^{rd}$ International Conference on Machine Learning*, Seoul, South Korea. PMLR 306, 2026. Copyright 2026 by the author(s).

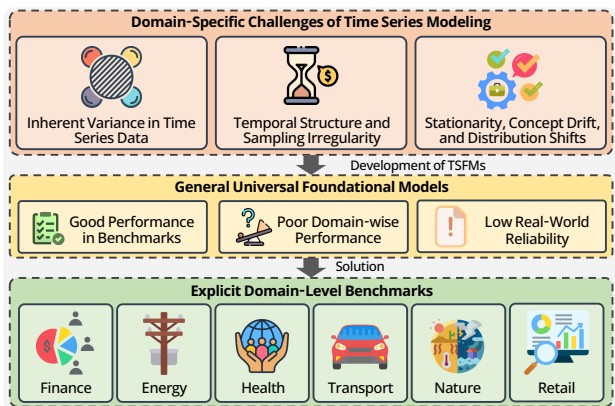

*Figure 1.* **Overview of our position.** Domain-specific differences in time series structure limit universal TSFMs, producing strong benchmark averages but inconsistent domain-wise reliability. We therefore argue for explicit domain-level benchmarks to expose transfer failures and enable meaningful evaluation.

## 1. Introduction

Inspired by breakthroughs in natural language processing and computer vision, where models like GPT (Brown et al., 2020), CLIP(Radford et al., 2021), and Gemini(Team et al., 2023) demonstrate remarkable zero-shot capabilities across diverse tasks, the machine learning community has begun pursuing **Time Series Foundation Models (TSFMs)** (Das et al., 2024b; Ansari et al., 2024) as a parallel paradigm. The central hypothesis is that a single, general-purpose model can be pretrained on large, diverse collections of time series to learn reusable temporal patterns. These patterns, such as seasonality, regime shifts, sparse events, and noise structures, appear consistently across different domains. To achieve this goal, TSFMs are trained on time series from finance, healthcare, mobility, and IoT sensors, aiming to provide the same benefits as language models: zero-shot forecasting, few-shot adaptation, and deployment without extensive feature engineering.

Several influential models have been developed following this vision, differing in their time series representations and scaling strategies while sharing a *"pre-train once, use broadly"* philosophy. TimesFM (Das et al., 2024b) adopts a GPT-style decoder-only Transformer architecture (≈200M parameters) and is trained on large corpora (100B time points) for direct sequence modeling at scale.

Chronos (Ansari et al., 2024) takes a different approach by adapting language modeling techniques, discretizing continuous time series values into tokens to support probabilistic forecasting with models ranging from 20M to 710M parameters. Building on tokenization strategies, Moirai (Woo et al., 2024b) introduces patch-wise tokenization to improve efficiency and context coverage, training on billions of points. Time-MoE (Shi et al., 2025) pushes scaling further with 2.4B parameters via sparse expert routing. Other approaches include Lag-Llama (Rasul et al., 2023), which emphasizes probabilistic forecasting, and MOMENT (Goswami et al., 2024), which adapts a T5-like objective through masked time-series modeling. Despite these architectural and tokenization choices, all models converge on a shared principle: large-scale cross-domain pretraining acts as a universal temporal prior, allowing a single model to generalize across datasets and forecasting tasks with minimal per-application redesign.

However, time series data exhibit fundamentally different statistical and structural properties across domains, challenging the assumption that a single universal prior can transfer reliably. In healthcare, observations are often irregularly sampled and multi-rate. For example, high-frequency signals such as ECG may arrive at millisecond resolution, while laboratory measurements are recorded hourly or even daily, creating temporal structures that are difficult to reconcile with fixed-interval tokenization and patching (Li et al., 2025a; Rubanova et al., 2019; Horn et al., 2020). Moreover, clinical data frequently contain informative missingness, where the absence of a measurement itself carries diagnostic meaning, further complicating unified temporal encoding (Lipton et al., 2016). In finance, the dominant challenges are different: time series exhibit high volatility, near-random-walk behavior, and non-stationarity driven by regime shifts, market microstructure effects, and evolving participant behavior (Cont, 2001; Hamilton, 1989). High-frequency trading introduces event-driven sampling, violating uniform-interval assumptions altogether. More broadly, time series vary sharply in their generative mechanisms, predictability, temporal resolution, and signal-to-noise ratios. Energy patterns follow physical laws while retail demand reflects human behavior, atmospheric processes exhibit chaotic dynamics while utility loads remain stable, and temporal scales range from sub-second sensor measurements to monthly economic indicators. These differences indicate that time series domains often constitute distinct learning problems, not merely surface variations of a universal task (Gama et al., 2014; Lu et al., 2018).

To cope with this heterogeneity, TSFMs adopt architectural and preprocessing strategies to standardize inputs for a shared Transformer backbone. Patching divides series into fixed-length windows treated as tokens (Nie et al., 2023; Woo et al., 2024b), but assumes temporal continuity and breaks down under irregular sampling. Channel-independence models each dimension separately, accommodating varying input variables but may discard cross-variable dependencies important in certain applications such as healthcare (Woo et al., 2024a). Instance normalization enables scale-invariant cross-domain transfer (Zeng et al., 2023), yet erases meaningful magnitude information such as absolute blood pressure thresholds unless reversible techniques are applied. Self-attention captures long-range dependencies, but its effectiveness depends on upstream tokenization choices. These design choices encode domain-specific inductive biases that fail outside their intended regimes. Patching acts as a low-pass filter unsuitable for high-frequency signals, while channel-independence may break down when cross-channel interactions are causal. Most critically, cross-domain pretraining induces gradient conflicts (Kouw & Loog, 2019): batches mixing medical series (requiring irregular timing sensitivity) with financial series (requiring regime-change robustness) produce conflicting updates that improve one domain while degrading another. The resulting representations reflect a compromise across incompatible objectives rather than a genuinely transferable temporal prior.

Motivated by this tension, **we argue that TSFMs require explicit domain-level benchmarks for meaningful assessment and deployment, since aggregated global leaderboards obscure domain-specific failure modes and produce misleading signals about real-world utility across diverse data-generating processes.** In this position paper, we explain why performance inconsistency emerges by linking it to domain-specific properties such as irregular sampling, non-stationarity, and heterogeneous noise that violate the assumptions embedded in current TSFM architectures and pretraining pipelines (§2). We also evaluate seven state-of-the-art TSFMs: TimesFM, Chronos, Moirai, Time-MoE, TimeGPT, Toto, and Sundial, across 72 datasets spanning six domains (healthcare, finance, energy, nature, transport, and retail), using consistent evaluation protocols for point forecasting accuracy (MSE, MAE) and ranking consistency (Demšar, 2006), showing that performance leadership remains fragmented rather than stable across domains (§4). Our findings call for domain-stratified benchmarks that surface application-specific strengths and weaknesses, enabling practitioners to select models with confidence and guiding the development of adaptive, hybrid TSFM designs that generalize reliably under real-world domain shift.

## 2. Domain-Specific Challenges of TSFMs

In this section, we explore the structural properties of time series data that create challenges for universal models, providing a foundation for understanding when domain-specific approaches may be preferable. We do not argue that univer-

sal models fail entirely, but rather that understanding these domain-specific utility is essential for practitioners deciding between universal and specialized approaches.

## 2.1. Inherent Variance in Time Series Data

Here, we explore the inherent variance in time series data that challenges the premise of a single universal temporal prior and motivates our position. Time series arising from physical, biological, and economic systems are governed by fundamentally different mechanisms, noise processes, and predictability horizons. For instance, an EEG voltage trace, an ICU vital-sign trajectory, and an equity return series emerge from distinct feedback loops and constraints. As a result, inductive biases that enable accurate forecasting in one domain often fail to transfer to others.

**Presence of Deterministic Laws.**

Many high-value forecasting domains are governed by deterministic laws with chaotic sensitivity. Large-scale atmospheric dynamics, for example, are described (under certain approximations) by the incompressible Navier–Stokes equations(Bistafa, 2024):

$$\frac{\partial \mathbf{u}}{\partial t} + (\mathbf{u} \cdot \nabla)\mathbf{u} = -\frac{1}{\rho}\nabla p + \nu \nabla^2 \mathbf{u} + \mathbf{f}, \qquad (1)$$

where $\mathbf{u}$ is velocity, $p$ pressure, $\rho$ density, $\nu$ viscosity, and $\mathbf{f}$ denotes external forcing. The shared physics across regions creates repeatable structure, enabling pre-training on weather or climate data to transfer smooth dynamical features across geographies and seasons. In contrast, financial markets are modeled as stochastic and partially adversarial environments, where exploitable patterns diminish through trading feedback. A canonical baseline is Geometric Brownian Motion (GBM), which models asset prices via the stochastic differential equation: $dS_t = \mu S_t \, dt + \sigma S_t \, dW_t$, where $\mu$ is the drift, $\sigma$ the volatility, and $W_t$ a Wiener process (Black & Scholes, 1973). Under strong forms of the Efficient Market Hypothesis (Fama, 1970), discounted prices behave as approximate martingales, making the current value a competitive forecast baseline and limiting persistent predictive structure. Because of this fundamental difference, aggregated benchmarks across domains can obscure domain-specific failures, making explicit domain-level evaluation essential for assessing transfer.

**Statistical Markers..** Several statistical markers further highlight how incompatible domains can be. The Hurst exponent $H$ summarizes long-range dependence: $H > 0.5$ indicates persistent behavior, $H = 0.5$ corresponds to random-walk-like dynamics, and $H < 0.5$ indicates mean reversion. Geophysical and hydrological series often exhibit $H > 0.5$, reflecting long-memory variability, whereas liquid financial markets are frequently closer to $H \approx 0.5$, consistent with near-martingale behavior (Hurst, 1951; Lo, 1991). Signal-to-noise ratio (SNR) also separates regimes: physics-driven domains often exhibit higher effective SNR, while high-frequency financial returns are dominated by noise, heavy tails, and regime shifts, making robust pattern extraction substantially harder (Mandelbrot & Hudson, 2007). Finally, positive Lyapunov exponents in chaotic systems bound predictability horizons, implying an intrinsic limit on how far ahead accurate forecasts are possible even with perfect models (Lorenz, 1963).

This variance has direct implications for TSFM pre-training. When a model is optimized over a heterogeneous corpus $\mathcal{D} = \bigcup_i \mathcal{D}_i$, gradients arising from different domains can conflict (Yu et al., 2020). In particular, domain-specific updates such as $\nabla_\theta \mathcal{L}_{\text{weather}}$ and $\nabla_\theta \mathcal{L}_{\text{finance}}$ may point in nearly orthogonal or opposing directions:

$$\langle \nabla_\theta \mathcal{L}_{\text{weather}}, \nabla_\theta \mathcal{L}_{\text{finance}} \rangle \approx 0 \quad \text{or} \quad < 0. \qquad (2)$$

Such conflicts imply that improving performance in one generative regime can degrade another, encouraging convergence toward a statistical average that is suboptimal for both physics-driven and stochastic domains. This failure mode is structural, not merely a limitation of data scale or compute. It follows that claims of universal TSFM capability must be tested under explicit domain stratification, since aggregated benchmarks can obscure sharp domain-level inconsistencies. Domain-aware evaluation is therefore a prerequisite for understanding when universal pre-training helps, when it induces negative transfer, and when domain-conditioned or specialized foundations are required.

## 2.2. Temporal Structure and Sampling Irregularity

Time series are defined not only by observed values but also by the temporal process that generates observations. This temporal structure is strongly domain-dependent: domains encode meaning in both *what* is measured and *when* it is measured, so time assumptions that hold in one setting can fail in another, placing intrinsic limits on universal models.

**Variance in Temporal Structure**. Most forecasting pipelines and tokenization-based TSFMs implicitly assume a uniform time grid, $t_i = t_0 + ic$, equivalently enforcing constant spacing $\Delta t_i = t_{i+1} - t_i = c$ (e.g., 15 minutes or 1 hour). In practice, timestamps are often produced by an explicit *observation-time process*, and sampling gaps can be irregular, bursty, and history-dependent. A convenient abstraction models observation times as a point process: $t_1, t_2, \cdots \sim \text{PP}(\lambda_\mathcal{D}(t \mid \mathcal{H}_t))$, where $\mathcal{H}_t$ is the event history and $\lambda_\mathcal{D}$ captures domain-specific sampling mechanisms (Rubanova et al., 2019). The induced gaps, $\Delta t_i = t_{i+1} - t_i$, therefore, vary across domains and are frequently informative rather than random, especially in clinical settings (Lipton et al., 2016). This creates a temporal distribution shift: models trained under regular-time assumptions treat index distance as time distance, an inductive mismatch that

degrades cross-domain generalization.

**Time Series Regularity Spectrum.** We characterize domains along a *regularity spectrum* (Figure 2). Weather and energy often exhibit *homogeneous sampling* under standardized protocols (e.g., smart meters at 15-minute intervals), making fixed-length patch tokenization generally effective (Woo et al., 2024b; Rasul et al., 2023). In contrast, healthcare and high-frequency finance show *heterogeneous sampling* with highly variable, often event-driven observation patterns. In clinical data, ECG signals may be recorded at millisecond intervals while laboratory tests are conducted hours or days apart based on clinical judgment (Li et al., 2025a), a multi-rate structure that fundamentally challenges uniform-interval tokenization.

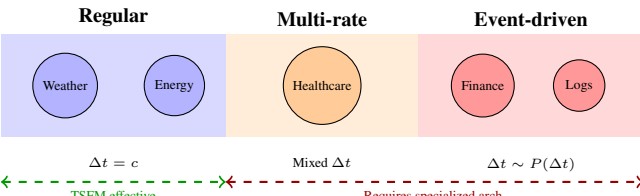

*Figure 2.* **Time Series Regularity Spectrum.** Sampling regimes vary across domains. Fixed patching works in regular settings, while multi-rate and event-driven data require time-aware or continuous-time biases.

This mismatch is particularly acute in ICU settings, where different physiological signals operate on vastly different timescales. ECG monitoring captures continuous waveforms at 100-500 Hz, while laboratory assays (e.g., blood chemistry panels) are ordered hours or days apart. These streams are rarely aligned and exhibit *informative missingness*, where the absence of a test reflects clinical judgment rather than random dropout (Lipton et al., 2016; Che et al., 2018). A TSFM trained primarily on regular sampling may treat such gaps as missing values to impute, potentially erasing semantically meaningful absence patterns. Fixed-length patching assumes uniform temporal intervals; when observation gaps vary from milliseconds to days, patch boundaries become arbitrary and fail to capture multi-rate temporal dynamics (Li et al., 2025a). This requires architectural redesign beyond standard patch-based approaches.

**Spectral Content in Time Series Modeling**. Domains also differ in *spectral content*. The Fourier transform,

$$X(f) = \int_{-\infty}^{\infty} x(t) \, e^{-2\pi i f t} \, dt, \tag{3}$$

highlights that macroeconomic indicators are dominated by low frequencies, while vibration and audio sensors contain high-frequency components. Deep networks exhibit *spectral bias*, learning low-frequency structure first and converging more slowly on high-frequency features (Rahaman et al., 2019); frequency-aware time series architectures have been proposed to mitigate this effect (Kang et al., 2024). As a result, TSFMs pretrained on low-frequency domains

can behave like *low-pass filters* when transferred to high-frequency sensors, smoothing the anomalies and transients they should detect. This spectral mismatch induces negative transfer that fine-tuning alone may not fully correct, since learned representations remain biased toward the frequency bands emphasized during pretraining.

### 2.3. Stationarity, Concept Drift, and Distribution Shifts

**Stationarity**. Generalization typically assumes that the training distribution $P_{\text{train}}$ approximates the test distribution $P_{\text{test}}$. In time series, this assumption is routinely violated by *non-stationarity*, where statistical properties evolve over time. Critically, the form and severity of non-stationarity vary sharply across domains, so a single universal strategy for distribution shift is inherently misaligned with real-world forecasting and motivates our position that TSFMs require explicit domain-level benchmarks.

A series $\{X_t\}$ is *weakly stationary* if its mean $\mathbb{E}[X_t] = \mu$ and variance $\text{Var}(X_t) = \sigma^2$ are constant and its autocovariance $\gamma(h) = \text{Cov}(X_t, X_{t+h})$ depends only on lag $h$. Some domains partially approximate this behavior, such as energy loads with stable seasonal structure under a slowly varying envelope, but others violate it fundamentally. In finance, forecasting is often framed in terms of *log-returns*. Let $S(t)$ be the asset price and $X(t) = \ln S(t)$ its log-price; the return over horizon $\Delta t$ is $r(t, \Delta t) = X(t + \Delta t) - X(t)$. Although $X(t)$ is frequently idealized as a random walk, empirical returns are far from i.i.d. Gaussian: linear autocorrelations are often weak, volatility is persistent (clustering), and returns are heavy-tailed (Cont, 2001). A TSFM pretrained on strongly periodic domains (e.g., energy) may therefore over-weight seasonality as a universal cue and misallocate capacity in finance, where time-varying volatility and shocks dominate.

**Concept Drift and Distribution Shifts**.

Beyond marginal changes, domains differ in whether the conditional relationship $P(Y|X)$ remains stable (Gama et al., 2014; Quiñonero-Candela et al., 2009). Under *covariate shift*, $P(X)$ changes while $P(Y|X)$ is approximately invariant; for example, climate change may shift temperature distributions while pressure and wind physics persist. Under *concept drift*, $P(Y|X)$ itself evolves, as in finance, where indicator and return relationships change across regimes (Hamilton, 1989). The 2008 financial crisis illustrates this risk: long-standing cross-asset correlations inverted abruptly, rendering pre-crisis models not only inaccurate but overconfident.

Covariate shift and concept drift are both forms of dataset *distribution shift*, which occurs when training and deployment distributions diverge (Quiñonero-Candela et al., 2009). These shifts vary systematically across domains. Finance

exhibits abrupt concept drift requiring frequent updates; energy shows predictable seasonal covariate shift suited to periodic recalibration; healthcare presents gradual patient-specific drift that challenges standard fine-tuning (Lu et al., 2019). A universal model cannot simultaneously trust stable physical relationships and remain skeptical of historically predictive patterns.

When TSFMs are pretrained on an aggregated corpus $\mathcal{D} = \bigcup_i \mathcal{D}_i$, they receive conflicting optimization signals, as gradients across domains are often nearly orthogonal. The resulting representation becomes a statistical average that lacks skepticism for rapid regime switches, adaptivity for patient-specific drift, and appropriate calibration when periodic domains dominate training. Fine-tuning cannot fully resolve this mismatch without catastrophic forgetting (Kouw & Loog, 2019), reinforcing that domain-aware evaluation is necessary for reliable deployment.

**Takeaway.** Together, these domain-specific challenges provide a theoretical basis for our central position that TSFMs do not yet behave as universal forecasters. The effective hypothesis class for forecasting varies across domains in causal structure, sampling process, spectral content, and drift dynamics, forcing a shared representation to absorb incompatible inductive biases. When pretraining aggregates these regimes, optimization becomes structurally conflicted, as gradients from physics-driven persistence, event-driven irregularity, and stochastic regime switching compete for the same parameters and produce statistically averaged representations rather than domain-optimal ones. These cross-domain failures are therefore not simply artifacts of limited scale or compute, but predictable outcomes of heterogeneous data-generating processes and distribution shift.

In the next section, we empirically validate our position by benchmarking leading TSFMs across six domains under a unified protocol, quantifying fragmentation in both accuracy and ranking stability.

## 3. Why Existing TSFM Benchmarks Fail?

Evaluating time series foundation models requires diverse benchmark suites that span multiple application domains. The three most prominent benchmarks such GIFT-Eval (Aksu et al., 2024), Monash (Godahewa et al., 2021d), and TSFM-Bench (Li et al., 2025b) collectively provide hundreds of datasets across domains including finance, healthcare, energy, transportation, and retail. However, closer examination reveals substantial domain imbalance in their composition.

Table 1 presents the domain distribution within GIFT-Eval and TSFM-Bench. Green highlights indicate the highest percentage within each column; red highlights indicate the lowest. GIFT-Eval, with 144,246 time series across

*Table 1.* Benchmark Data Statistics Aggregated by Domain.

| Domain | GIFT-Eval | | TSFM-Bench | |
|---|---|---|---|---|
| | # Series (%) | # Obs (%) | # Series (%) | # Obs (%) |
| Economic/Finance | 100.0K (69.3%) | 25.3M (16.0%) | 236 (4.9%) | 238.8K (0.6%) |
| Energy | 2.0K (1.4%) | 74.1M (46.9%) | 493 (10.3%) | 17.0M (42.2%) |
| Nature | 32.6K (22.6%) | 3.2M (2.0%) | 54 (1.1%) | 1.9M (4.8%) |
| Healthcare | 1.0K (0.7%) | 129.4K (0.1%) | 955 (20.0%) | 1.3M (3.3%) |
| Sales | 3.7K (2.6%) | 671.7K (0.4%) | – | – |
| Traffic/Transport | 1.3K (0.9%) | 38.0M (24.1%) | 1.0K (21.6%) | 18.2M (45.2%) |
| Web | 3.5K (2.4%) | 16.6M (10.5%) | 2.0K (41.9%) | 1.6M (3.9%) |
| **Total** | **144.2K** | **158.0M** | **4.8K** | **40.2M** |

seven domains, is heavily skewed toward Economic/Finance (69.3% of series) and Nature (22.6%), leaving Healthcare with merely 0.7% representation and only 0.1% of total observations. TSFM-Bench exhibits different but equally problematic imbalance: Web data dominates at 41.9% of series, while Traffic/Transport contributes 45.2% of total observations. The benchmarks also show contrasting domain emphases: Economic /Finance represents just 0.6% of observations in the TSFM-Bench compared to 16.0% in GIFT-Eval, while Nature accounts for only 1.1% of the series in the TSFM-Bench versus 22.6% in GIFT-Eval. TSFM-Bench entirely omits Sales or Retail data. This inconsistent coverage means models evaluated on either benchmark alone may show inflated or deflated performance depending on their domain strengths. This domain imbalance motivates our stratified evaluation approach. Instead of aggregate scores, we group our 72 datasets into six balanced domains: Health, Finance, Nature, Transport, Retail, and Energy, so each contributes equally and enables fair cross-domain comparison across distinct data collection processes.

## 4. Empirical Validation

**Experimental Design.** We evaluate seven state-of-the-art TSFMs, namely TimesFM, Chronos, Moirai, Time-MoE, TimeGPT, Toto, and Sundial, across 72 datasets spanning six domains: healthcare, finance, energy, nature, transport, and retail. All models are benchmarked under a unified evaluation protocol, using point forecasting accuracy metrics (MSE and MAE) and ranking consistency following Demšar (2006). For fairness, each TSFM is evaluated in its strongest reported configuration, using the best-performing setting described in the original paper or official implementation. Additional details on the datasets, preprocessing, and implementation are provided in Appendix A.

### 4.1. Domain-wise Performance

Detailed domain-wise results are reported in Table 2 (energy) in the main paper, while Tables 5 (healthcare), 6 (finance), 7 (nature), 8 (transport), and 9 (retail) are provided in the appendix due to space constraints. On energy datasets (Table 2), TSFMs exhibit strong performance but still show fragmented leadership rather than a single consistently dominant model. Averaged across datasets

*Table 2.* Energy Time Series Forecasting Results

| Dataset | Horizon | Chronos | | Sundial | | TimeGPT | | TimesFM | | TimesMOE | | ToTo | | Moirai | |
|---|---|---|---|---|---|---|---|---|---|---|---|---|---|---|---|
| | | MSE | MAE | MSE | MAE | MSE | MAE | MSE | MAE | MSE | MAE | MSE | MAE | MSE | MAE |
| AUS Elec Demand | 24 | 0.0965 | 0.2824 | **0.0452** | **0.1824** | 0.1628 | 0.3390 | 0.2049 | 0.4283 | 0.0616 | 0.2238 | 0.2572 | 0.4547 | 0.5435 | 0.6183 |
| | 48 | 0.2468 | 0.3903 | **0.0300** | **0.1471** | 0.2701 | 0.4754 | 0.1051 | 0.2829 | 0.0403 | 0.1565 | 0.4696 | 0.5836 | 0.3484 | 0.5248 |
| | 96 | 0.0899 | 0.2104 | **0.0233** | **0.1254** | 0.6010 | 0.6545 | 0.0725 | 0.2145 | 0.0355 | 0.1301 | 0.2735 | 0.4344 | 0.3047 | 0.4379 |
| Electricity Weekly | 24 | 1.8938 | 0.8021 | 4.0820 | 1.8735 | 1.6757 | 0.9854 | 1.9812 | 0.7917 | 1.5199 | 0.9913 | **0.5357** | **0.3301** | 1.8180 | 0.8297 |
| | 48 | 1.0033 | 0.4997 | 1.9888 | 1.1821 | 0.9494 | 0.7653 | 0.9929 | 0.5020 | 0.8377 | 0.7212 | **0.5051** | **0.3309** | 0.9518 | 0.5685 |
| | 96 | 1.5384 | 0.7182 | 1.3030 | 0.7318 | 1.4077 | 0.7351 | 1.5888 | 0.7170 | 1.1375 | 0.7835 | **0.4735** | **0.3294** | 1.4167 | 0.7237 |
| ETTh1 | 24 | 0.2283 | 0.3865 | 0.2420 | 0.4044 | 0.1807 | 0.3444 | 0.3594 | 0.4581 | **0.1646** | **0.3250** | 3.0815 | 1.4634 | 0.4355 | 0.4844 |
| | 48 | **0.2586** | **0.4203** | 0.2867 | 0.4248 | 0.4537 | 0.4996 | 0.5055 | 0.5251 | 0.4087 | 0.4860 | 2.7232 | 1.3733 | 0.5730 | 0.5773 |
| | 96 | 0.4905 | 0.5491 | **0.2191** | **0.3884** | 0.3825 | 0.5045 | 0.2465 | **0.3595** | 0.2792 | 0.4068 | 2.4586 | 1.3231 | 0.3683 | 0.4273 |
| ETTh2 | 24 | 0.3939 | 0.5651 | 0.4844 | 0.6033 | 0.2440 | 0.4401 | 0.2169 | 0.4138 | **0.1788** | **0.3762** | 0.3397 | 0.5180 | 0.2314 | 0.4050 |
| | 48 | 0.4504 | 0.5808 | **0.1561** | **0.3359** | 0.3284 | 0.4760 | 0.3249 | 0.4832 | 0.2606 | 0.4265 | 0.5246 | 0.6336 | 0.5369 | 0.6288 |
| | 96 | 0.3507 | 0.4583 | 0.6701 | 0.7525 | **0.2092** | **0.3461** | 0.2492 | 0.3887 | 0.2126 | 0.3512 | 0.7067 | 0.7373 | 0.4182 | 0.5028 |
| ETTm1 | 24 | 0.4960 | 0.5354 | 0.3141 | 0.4137 | 0.5773 | 0.6387 | **0.2163** | **0.3580** | 0.2348 | 0.3806 | 0.6619 | 0.6351 | 0.7383 | 0.7413 |
| | 48 | 1.1027 | 0.8624 | 0.4640 | 0.4702 | 0.6325 | 0.6872 | 0.3541 | 0.4923 | **0.2976** | **0.4418** | 0.9816 | 0.8668 | 1.9918 | 1.2635 |
| | 96 | 2.3457 | 1.0366 | 0.2773 | 0.3990 | 1.2350 | 0.9271 | **0.1379** | **0.2574** | 0.1871 | 0.3144 | 0.8213 | 0.6837 | 2.9523 | 1.2133 |
| ETTm2 | 24 | **0.0696** | 0.2330 | 0.2464 | 0.4649 | 0.0910 | 0.2769 | 0.1077 | 0.2769 | 0.1497 | 0.3213 | 0.0811 | **0.2314** | 0.1746 | 0.3590 |
| | 48 | 0.8794 | 0.8979 | 0.8366 | 0.7664 | 1.2731 | 1.0895 | **0.2987** | **0.4924** | 0.4734 | 0.6398 | 0.7530 | 0.8361 | 1.1056 | 0.9997 |
| | 96 | 0.3029 | 0.4796 | **0.2179** | **0.3873** | 0.3614 | 0.5333 | 0.2192 | 0.3955 | 0.2420 | 0.4093 | 0.2821 | 0.4785 | 0.3386 | 0.4840 |
| London SmartMeters | 24 | 2.3292 | **0.9403** | 2.8266 | 1.1401 | 3.5515 | 1.3026 | 2.7089 | 1.0717 | **1.9351** | 0.9475 | 5.8690 | 1.9358 | 2.6385 | 1.1145 |
| | 48 | 2.3505 | 0.9762 | **1.7465** | **0.8874** | 2.6237 | 1.0995 | 2.3544 | 0.9456 | 2.0048 | 0.8904 | 7.0594 | 2.0329 | 2.3622 | 0.9592 |
| | 96 | 1.1737 | 0.6105 | 1.2863 | 0.6569 | 1.5539 | 0.8628 | 1.2064 | 0.6105 | **1.0528** | **0.6025** | 6.7537 | 1.9636 | 1.3359 | 0.6795 |
| Solar 10min | 24 | **0.0001** | 0.0230 | 0.0034 | 0.0038 | 0.0087 | 0.0752 | 0.0008 | 0.0258 | 0.0041 | 0.0105 | 0.0087 | **0.0007** | 0.0069 | 0.0027 |
| | 48 | 0.1077 | 0.3185 | 0.0035 | 0.0310 | 0.1179 | 0.3414 | 0.0227 | 0.0966 | 0.0114 | 0.0791 | **0.0003** | **0.0103** | 0.1430 | 0.3469 |
| | 96 | 0.2472 | 0.4656 | 0.7006 | 0.5764 | **0.1308** | **0.3231** | 0.5050 | 0.4818 | 0.5495 | 0.5748 | 0.2578 | 0.4299 | 0.2039 | 0.4181 |
| Average | | 0.7686 | 0.5518 | 0.7689 | 0.5562 | 0.7926 | 0.6134 | 0.6242 | 0.4612 | 0.5116 | 0.4579 | 1.4950 | 0.7757 | 0.9141 | 0.6379 |
| 1st Count | | 3 | 2 | 7 | 6 | 2 | 2 | 3 | 4 | 5 | 4 | 4 | 6 | 0 | 0 |

and horizons, TimesMoE achieves the best overall accuracy (MSE 0.5116, MAE 0.4579), outperforming TimesFM (MSE 0.6242, MAE 0.4612) and Chronos (MSE 0.7686, MAE 0.5518). However, wins are distributed across models: Sundial leads most often with 7 first-place finishes in MSE and 6 in MAE, while TimesMoE records 5 MSE wins and 4 MAE wins, indicating no universally stable leader even within a single domain. As additional experiments, the results for the two new models, TabPFN-TS and Moment (detailed in §B.2), are provided in the appendix and are not included in the overall ranking. For MOMENT, we evaluated it on the all domain, it performs well in Retail and Nature, matching top models on structured sales and climate data. TabPFN-TS demonstrates clear strengths in Health and Energy, securing the most first-place results in Health on clinical, hospital, and demographic datasets. These patterns extend across the remaining domains, reinforcing our position that TSFMs do not yet behave as universal forecasters and must be evaluated with explicit domain-level benchmarks.

## 4.2. Aggregated Performance and Ranking

Across domains, Figure 3 shows that TSFM rankings are highly non-uniform and rarely stable across application areas. No model stays consistently top-ranked across healthcare, finance, nature, transport, retail, and energy, and rank reversals are common. TimesFM performs strongly in retail but drops in finance and nature, while ToTo shows the opposite trend with comparatively better ranks in finance and

nature but weaker performance elsewhere. Even competitive models such as Chronos and TimesMOE exhibit wide spreads, indicating sensitivity to dataset choice and horizon within each domain. Overall, these distributions show that global averages can hide frequent domain-specific failures, supporting our position that aggregated leaderboards are unreliable indicators of deployment utility.

Figure 4 further confirms this fragmentation by revealing distinct domain performance footprints for each model. Rather than uniform strength, TSFMs show peaked behavior, with advantages concentrated in a few domains and clear weaknesses in others. TimesFM exhibits the most pronounced skew, with a strong retail advantage that does not transfer, while ToTo peaks in finance and nature but remains weaker in transport and retail. Sundial and Moirai show narrower gains, reinforcing that inductive biases tuned to particular temporal regimes do not generalize broadly. Together, these patterns affirm that TSFMs lack cross-domain consistency and motivate explicit domain-level benchmarks to evaluate transfer reliably.

## 4.3. Domain-Specific Performance Differences

Table 3 summarizes the Kruskal-Wallis test (Kruskal & Wallis, 1952) (detailed in §C) outcomes at horizon $h = 24$.

For both MAE and MSE, most models yield statistically significant evidence against the null hypothesis of domain-invariant performance, indicating that error distributions vary systematically across application domains rather than

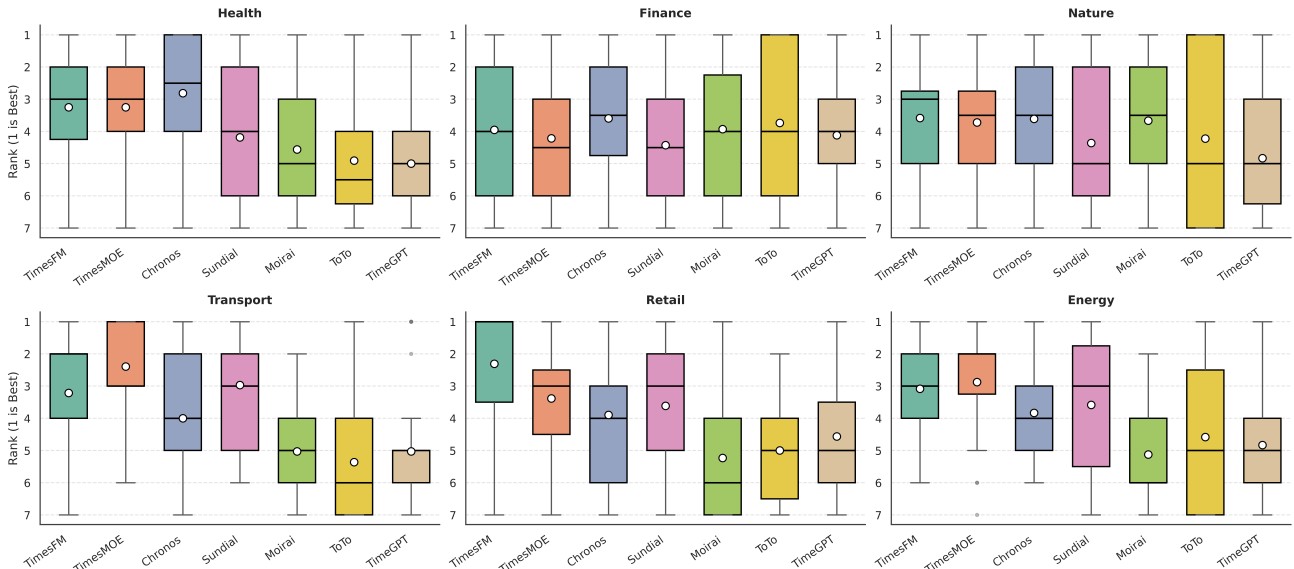

*Figure 3.* **Model Rank Distribution (MAE) Across Domains.** Box plots show the distribution of model rankings within each domain, where rank 1 indicates best performance. White circles denote mean rank; horizontal lines indicate median rank.

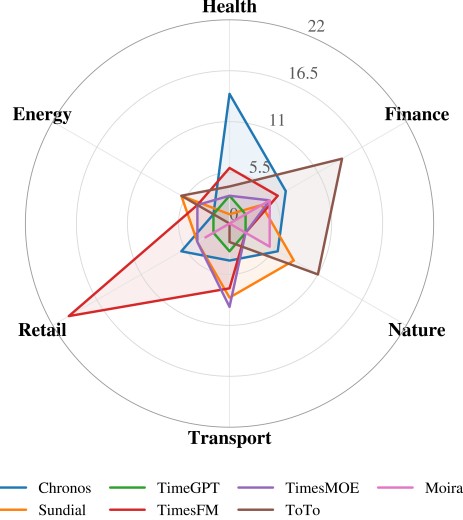

*Figure 4.* Domain-Wise Win Counts of Time-Series Foundation Models (MAE) (See Figure 6 for MSE rankings.)

remaining stable under a single global profile. The domain effect is generally stronger under MSE, where several models attain $p \leq 0.01$ (or better), suggesting that cross-domain shifts disproportionately amplify squared-error sensitivity through occasional high-magnitude deviations. MAE also shows consistent significance across models but at comparatively weaker levels, which aligns with its greater robustness to heavy-tailed errors and outlier behavior. Overall, these results provide quantitative evidence that time-series foundation model performance is *domain-dependent* at a fixed horizon, motivating domain-aware benchmarking and reporting beyond aggregate evaluations. More details are available in §C.2.

*Table 3.* Global domain effect at Horizon $= 24$ using the Kruskal–Wallis test. Reported values are $p$-values with significance markers: $ns$ ($p > 0.05$), * ($p \leq 0.05$), ** ($p \leq 0.025$), *** ($p \leq 0.01$).

| Model | MAE ($p$, sig.) | MSE ($p$, sig.) |
|---|---|---|
| Chronos (Ansari et al., 2024) | 0.0444* | 0.0146** |
| Moirai (Woo et al., 2024b) | 0.0634 (*ns*) | 0.0123** |
| Sundial (Liu et al., 2025) | 0.0376* | 0.0213** |
| TimeGPT (Garza et al., 2024) | 0.0112** | 0.0047*** |
| TimesFM (Das et al., 2024a) | 0.0190** | 0.0110** |
| TimesMOE (Shi et al., 2025) | 0.0118** | 0.0048*** |
| ToTo (Cohen et al., 2024) | 0.0201** | 0.0104** |

## 5. Discussion

Our results provide empirical support for the theoretical argument developed in Section 2: **current TSFMs do not yet behave as universal forecasters**, because cross-domain transfer is constrained by structural differences in data generation, sampling, and drift. The observed performance fragmentation is systematic rather than noise around a shared optimum. Consistent rank reversals across domains indicate incompatible inductive biases: stable, seasonal domains (e.g., energy) favor models capturing smooth long-range structure, while stochastic, regime-shifting domains (e.g., finance) reward robustness to heavy tails, volatility, and rapidly changing $P(Y \mid X)$. A single shared representation must therefore encode conflicting assumptions about predictability and stationarity.

Importantly, this inconsistency persists even when TSFMs are evaluated in their strongest reported configurations. In the energy domain (Table 2), the best average model is TimesMoE (MSE 0.5116, MAE 0.4579), yet Sundial achieves the largest number of first-place finishes (7 in MSE

and 6 in MAE), demonstrating that leadership is unstable even within a comparatively structured domain. This instability becomes more pronounced when moving to cross-domain evaluation: Figure 3 shows wide rank distributions and frequent reversals across healthcare, finance, nature, transport, retail, and energy, indicating that global mean performance is an unreliable proxy for domain-level utility. The domain footprint in Figure 4 further reinforces that models specialize implicitly, with concentrated strengths in a subset of domains and clear weaknesses elsewhere, rather than expressing uniformly transferable capability. Finally, our Kruskal-Wallis analysis (Table 3) provides statistical evidence that error distributions differ systematically by domain for most models, rejecting the hypothesis of domain-invariant performance under a single global profile.

These findings reaffirm our position that **TSFMs require explicit domain-level benchmarks**. Aggregated leaderboards can overstate progress by averaging across domains where a model is well-aligned with the underlying temporal structure, while concealing failure modes in domains where its assumptions break. In practice, what matters is not whether a TSFM performs well on average, but whether it is reliable within the target application domain, under the relevant sampling regime, noise structure, and drift dynamics. Our evidence suggests that without domain-stratified evaluation, the community risks mistaking benchmark coverage for universality, and practitioners risk deploying models whose apparent global strength does not translate into consistent domain-level performance.

## 6. Alternative Views

**Alternative View 1: Existing Benchmarks Already Provide Sufficient Granularity.** One might argue that per-dataset reporting in current benchmarks already offers sufficient diagnostic resolution, making explicit domain-level evaluation unnecessary. However, Table 1 shows that domain representation is highly uneven, with some domains contributing far more datasets and observations than others. Consequently, aggregate metrics are dominated by well-represented domains, while systematic failures in sparsely represented but high-stakes domains remain hidden. Per-dataset results alone lack the structure needed to assess whether models generalize consistently across domains rather than excelling where data are abundant, conflating coverage with competence.

**Alternative View 2: Universality Emerges Through Adaptation, Not Evaluation.** A related view holds that expressive TSFMs should achieve cross-domain robustness via fine-tuning or in-context learning (ICL), obviating domain-level benchmarks. While adaptation can mitigate mismatch in stable, homogeneous domains, it fails under heterogeneous pretraining: cross-domain optimization yields sta-

tistical averages rather than transferable priors, and fine-tuning often induces catastrophic forgetting (Yang et al., 2024). Likewise, ICL cannot fix inductive misalignment when base representations are flawed (Dong et al., 2024). In fast-evolving domains such as finance (Ang & Timmermann, 2012) and health, requirements change faster than prompting or lightweight adaptation can accommodate (Das et al., 2025). Evidence from expert-combination frameworks further indicates that robust forecasting depends on specialization rather than universal representations (Ni et al., 2024).

**Implication for Our Position.** These alternatives raise valid expectations about benchmarks and adaptation, but neither resolves the core limitation. Without domain-stratified benchmarks, apparent gains may reflect dominance by over-represented domains rather than genuine transfer. Adaptation does not remove this need; it depends on prior diagnosis of where and why base models fail. Explicit domain-level evaluation is therefore a prerequisite for assessing universality claims and determining when adaptation suffices versus when domain-specific modeling is required.

## 7. Call to Action

Our theoretical analysis and empirical findings support a paradigm shift: the community should report *domain-stratified* benchmarking instead of only reporting pooled global evaluation. Current practice masks domain-level failure modes behind aggregate metrics, preventing diagnosis of where TSFMs succeed versus fail. We identify three priorities, ordered by immediacy of impact.

**A1. Domain-Stratified Evaluation**. It is the lowest-cost, highest-impact change. Existing benchmarks like Monash(Godahewa et al., 2021d), GIFT-Eval (Aksu et al., 2024), and TSFM-Bench (Li et al., 2025b) already span multiple domains; hence, the community also need to report results *by domain* rather than as pooled averages. Our analysis shows that models ranking first overall can rank last on specific domains; without stratified reporting, selecting an appropriate TSFM for a given application becomes difficult for practitioners.

**A2. Evaluating Domain-Aware Models.** Domain-specific benchmarks would enable fair comparison of emerging domain-aware models against universal TSFMs. Specialized models like MIRA (Li et al., 2025a) (targeting medical irregularity and missingness) and FinCast (Zhu et al., 2025) (addressing financial non-stationarity) report strong gains on their target domains but lack standardized evaluation against general-purpose alternatives. Stratified benchmarks would reveal whether domain-aware components consistently outperform global models within their intended domains, guiding future architectural investment.

**A3. Cross-Domain Transfer Framework.** TSFMs need a principled framework for cross-domain transfer, such as one that predicts which source domains benefit which targets. Currently, practitioners cannot know in advance whether a model pretrained mostly on energy data will help or harm medical forecasting. Such a framework would characterize domain similarity (e.g., by sampling frequency, stationarity, or causal structure) and identify when negative transfer is likely, enabling more informed model selection rather than repeated domain-specific trial-and-error tests (Wang et al., 2018; Kouw & Loog, 2019).

## Software and Data

All datasets used in our experiments are publicly available benchmark datasets, and no proprietary or private data were used. Detailed descriptions of the datasets, preprocessing steps, and experimental setup are provided in the appendix. The time-series foundation models evaluated in this work are also publicly available through their official repositories or model releases. We use these public resources only for research and benchmarking purposes, following their respective licenses and usage terms.

## Acknowledgements

We thank the authors and maintainers of the time-series foundation models and benchmark suites used in this study for making their code, models, and datasets publicly available. We also acknowledge the broader time-series forecasting community for developing the resources that made this comparative analysis possible. This work was supported in part by the computational resources available to the authors. We are also grateful to the reviewers for their constructive feedback and thoughtful suggestions, which helped improve the clarity and presentation of this work.

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

# A. Extended Experimental Details

## A.1. Dataset Description

We evaluate TSFMs on 72 datasets spanning six domains: Healthcare, Finance, Energy, Nature, Transport, and Retail. To ensure diversity and reliability, our datasets are sourced from established benchmarks and trusted repositories, including Monash Time Series Forecasting Archive (Godahewa et al., 2021d), GIFT-Eval (Aksu et al., 2024), TSFM-Bench (Li et al., 2025b), Kaggle competitions, the UCI Machine Learning Repository, and Zenodo. This multi-source approach maximizes coverage across temporal granularities, domain-specific patterns, and real-world forecasting challenges. Below we describe the datasets used in each domain.

### A.1.1. HEALTHCARE

The healthcare domain comprises 11 datasets spanning diverse medical and epidemiological time series, including **COVID-19** (Li et al., 2025b) related intensive care admissions, mortality, and weekly case counts. **Hospital** records daily patient admission counts (Godahewa et al., 2021d). **Chickenpox** contains weekly chickenpox case counts from Hungary (Hun, 2021). **Illness** represents the percentage of patients with influenza-like illness (Godahewa et al., 2021d). **T1–T4** are heart rate time series from the MIT-BIH Arrhythmia Database (Moody & Mark, 2001). **US-Births** tracks daily birth records in the United States (Godahewa et al., 2021d).

### A.1.2. FINANCE

The finance domain includes 14 datasets representing stock prices and financial indicators. Eleven individual stock time series (**AAL**, **AAPL**, **BLDP**, **CASH**, **CBD**, **DLR**, **EDUC**, **GWW**, **HAS**, **NSC**, **PHG**) are sourced from public financial data repositories (Li et al., 2025b). **CIF2016** originates from the Computational Intelligence in Forecasting competition (Štěpnička & Burda, 2015). **NN5-Daily** and **NN5-Weekly** contain ATM cash withdrawal data at daily and weekly granularities (Li et al., 2025b).

### A.1.3. ENERGY

The energy domain consists of 8 datasets covering electricity consumption and generation patterns. **ETTh1**, **ETTh2**, **ETTm1**, and **ETTm2** provide hourly and 15-minute Electricity Transformer Temperature measurements (Li et al., 2025b). **AUS Elec Demand** tracks Australian electricity demand (Godahewa et al., 2021c). **Electricity Weekly** contains aggregated weekly electricity consumption (Godahewa et al., 2020a). **London SmartMeters** records household energy usage from UK smart meters (Godahewa et al., 2020b). **Solar 10min** captures solar power generation at 10-minute intervals (Godahewa et al., 2020d).

### A.1.4. NATURE

The nature domain encompasses 12 datasets related to environmental and climate measurements. **Indian-Climate-AQI**, **Indian-Climate-Humidity**, **Indian-Climate-Temperature**, and **Indian-Climate-Wind** provide air quality and meteorological measurements from India.[1] **Jena-Climate-H2OC** and **Jena-Climate-RH** capture water vapor concentration and relative humidity from Jena, Germany (Wu et al., 2021). **Ozone**, **Pollution**, **Saugeen-Day**, **Sunspot**, **Temp-Hourly**, and **Weather** cover atmospheric ozone concentration, air pollution, daily river flow, solar activity, and hourly temperature and weather observations (Godahewa et al., 2021d).

### A.1.5. TRANSPORT

The transport domain contains 11 datasets capturing traffic and mobility patterns. **Cross-1** through **Cross-6** track traffic at different road crossings (Axenie & Bortoli, 2020). **Pedestrian-Counts** records pedestrian traffic from sensors in Melbourne (Godahewa et al., 2020c). **Rideshare** captures rideshare demand data (Godahewa et al., 2021a). **Traffic-Hourly** and **Traffic-Weekly** provide San Francisco highway traffic volume at hourly and weekly temporal resolutions (Godahewa et al., 2020e;f). **Vehicle-Trips** records vehicle trip counts (Godahewa et al., 2021b).

---

[1] https://www.kaggle.com/datasets/ankushnarwade/indian-climate-dataset-20242025

A.1.6. RETAIL

The retail domain includes 12 datasets representing product sales and demand. **Amazon Sales**[2] and **Coffee Sales**[3] track e-commerce and retail transactions. **Online Retail** is a transactional dataset containing all transactions for a UK-based non-store online retailer (Chen, 2015). **Dominick's Dataset** contains retail scanner data (Godahewa et al., 2021d). **Rossmann** contains daily sales from Rossmann drug stores.[4] **Store 1 Item 1–Store 1 Item 5** and **Product 1–Product 3** capture item-level demand across retail locations.[5].

## A.2. Implementation Details

A.2.1. EXPERIMENTAL SCOPE

All experiments are conducted under a **univariate forecasting** setting. Each time series is treated independently, without leveraging cross-series or multivariate dependencies. This choice ensures a fair comparison across TSFMs with different architectural assumptions regarding input dimensionality.

A.2.2. DATA PREPROCESSING

We apply a standardized preprocessing pipeline across all datasets. Each time series is reformatted to contain a `timestamp` column and a `value` column, ensuring compatibility across all evaluated TSFMs. Missing values are imputed using domain-appropriate methods: for finance, we apply forward-fill followed by backward-fill for leading gaps; for healthcare, we use forward-fill while preserving missingness indicators where clinically relevant; for energy and nature domains, we apply linear interpolation for short gaps and forward-fill for isolated missing points. We then apply instance-wise z-score normalization before model inference:

$$\hat{x}_t = \frac{x_t - \mu}{\sigma}, \quad \text{where} \quad \mu = \frac{1}{T}\sum_{t=1}^{T} x_t, \quad \sigma = \sqrt{\frac{1}{T}\sum_{t=1}^{T}(x_t - \mu)^2} \tag{4}$$

where $x_t$ denotes the original value at time $t$, $\hat{x}_t$ is the normalized value, and $T$ is the length of the time series. Predictions are denormalized using $x_t = \hat{x}_t \cdot \sigma + \mu$ before computing evaluation metrics. Finally, a chronological train/test split is used, with the test set comprising the final observations corresponding to the forecast horizon.

A.2.3. MODEL CONFIGURATIONS

We evaluate the publicly available variant of each TSFM to ensure maximum model capacity. All models are used in zero-shot mode with their default inference hyperparameters from the official implementations. Table 4 summarizes the model specifications and key inference parameters.

Table 4 reports the configurations of the time series foundation models evaluated in this study. For each model, we use the largest publicly available variant and perform inference in a zero-shot setting without task-specific fine-tuning. The table lists the total number of parameters, with activated parameters shown separately for mixture-of-experts models, enabling a direct comparison of model capacity. Maximum context length indicates the longest input sequence each model can process, which varies substantially across models and directly affects their ability to capture long-range temporal dependencies. All models are evaluated under identical forecasting horizons of 24, 48, and 96 steps to ensure comparability.

The inference parameters column summarizes the default hyperparameters provided by the official implementations, reflecting realistic out-of-the-box usage. Models such as TimesFM and Moirai support substantially longer contexts, while Chronos, Toto, and Time-MoE operate with more constrained input lengths. Time-MoE stands out with the largest total parameter count, although only a subset of parameters is activated during inference. TimeGPT is included as a proprietary baseline, for which architectural details and model size are not publicly disclosed. Overall, the table highlights the diversity in model scale, context capacity, and inference design among contemporary time series foundation models.

---

[2]https://www.kaggle.com/datasets/rohiteng/amazon-sales-dataset
[3]https://www.kaggle.com/datasets/ihelon/coffee-sales
[4]https://www.kaggle.com/competitions/rossmann-store-sales/data
[5]https://www.kaggle.com/competitions/demand-forecasting-kernels-only/data

*Table 4.* TSFM Model Configurations. All models use their largest available variant in zero-shot mode. *Params* denotes total parameters (activated parameters in parentheses for MoE models). *Context* denotes maximum supported input length. Inference parameters show default settings used in our evaluation. [†]Proprietary model with undisclosed architecture.

| Model | Variant | Params | Max Context | Horizon | Inference Parameters |
|---|---|---|---|---|---|
| Chronos (Ansari et al., 2024) | 2 | 710M | 512 | 24, 48, 96 | `context_df,` `quantile_levels,` `future_df` |
| TimesFM (Das et al., 2024b) | 2.5 | 200M | 16,384 | 24, 48, 96 | `max_context,` `normalize_inputs,` `infer_is_positive` |
| Moirai (Woo et al., 2024b) | Large | 311M | 5000 | 24, 48, 96 | `prediction_length,` `windows, distance` |
| Time-MoE (Shi et al., 2025) | Large | 2.4B (1B) | 4096 | 24, 48, 96 | `normed_seqs,` `max_new_tokens` |
| Toto (Cohen et al., 2024) | Base | 151M | 512 | 24, 48, 96 | `prediction_length,` `num_samples,` `samples_per_batch` |
| Sundial (Liu et al., 2025) | Base | 128M | 2880 | 24, 48, 96 | `seqs, max_new_tokens,` `num_samples` |
| TimeGPT (Garza et al., 2024) | 1 | $-^†$ | $-^†$ | 24, 48, 96 | `df, h, level` |

### A.2.4. EVALUATION PROTOCOL

We evaluate all models at three forecast horizons: $h \in \{24, 48, 96\}$ time steps. Performance is measured using two standard metrics: Mean Squared Error (MSE) and Mean Absolute Error (MAE), computed on normalized predictions. All models are evaluated in a zero-shot setting without fine-tuning on the target datasets, reflecting practical deployment scenarios where labeled target data may be unavailable.

Mean Squared Error (MSE) is calculated as:

$$\text{MSE} = \frac{1}{T} \sum_{t=1}^{T} (y_t - \hat{y}_t)^2 \tag{5}$$

Mean Absolute Error (MAE) is calculated as:

$$\text{MAE} = \frac{1}{T} \sum_{t=1}^{T} |y_t - \hat{y}_t| \tag{6}$$

Here, $T$ denotes the total number of time steps in the evaluation window, $y_t$ represents the ground-truth value at time step $t$, and $\hat{y}_t$ denotes the corresponding model prediction. Both metrics are computed over normalized time series to ensure comparability across datasets and scales. MSE penalizes larger errors more strongly due to the squared term, while MAE provides a linear measure of average absolute deviation between predictions and true values.

# B. Extended Experimental Results

## B.1. Domain-wise Performance

Here we provide the detailed experimental results in Tables 5 (healthcare), 6 (finance), 7 (nature), 8 (transport), and 9 (retail).

*Table 5.* Health Time Series Forecasting Results

| Dataset | Horizon | Chronos | | Sundial | | TimeGPT | | TimesFM | | TimesMOE | | ToTo | | Moirai | |
|---|---|---|---|---|---|---|---|---|---|---|---|---|---|---|---|
| | | MSE | MAE | MSE | MAE | MSE | MAE | MSE | MAE | MSE | MAE | MSE | MAE | MSE | MAE |
| COVID-ICU | 24 | **0.0001** | **0.0066** | 0.0005 | 0.0194 | 0.0004 | 0.0166 | 0.0002 | 0.0077 | 0.0034 | 0.0473 | 0.0003 | 0.0139 | 0.0003 | 0.0082 |
| | 48 | **0.0005** | **0.0218** | 0.0167 | 0.1117 | 0.0091 | 0.0877 | 0.0031 | 0.0466 | 0.0051 | 0.0636 | 0.0073 | 0.0845 | 0.0048 | 0.0557 |
| | 96 | **0.0010** | **0.0281** | 0.0061 | 0.0570 | 0.0457 | 0.1918 | 0.0128 | 0.0870 | 0.0136 | 0.0954 | 0.0295 | 0.1682 | 0.0498 | 0.1943 |
| COVID-Deaths | 24 | 0.0041 | 0.0551 | 0.0351 | 0.1626 | **0.0016** | **0.0353** | 0.0057 | 0.0710 | 0.0163 | 0.0964 | 0.0104 | 0.0783 | 0.0399 | 0.1884 |
| | 48 | **0.0547** | **0.2172** | 1.439 | 1.145 | 0.3511 | 0.5438 | 0.0845 | 0.2803 | 0.0884 | 0.2720 | 0.4304 | 0.6340 | 0.0539 | 0.1981 |
| | 96 | 0.8522 | 0.7216 | 3.167 | 1.476 | 2.305 | 1.251 | 0.5924 | 0.6186 | **0.3685** | **0.4696** | 4.000 | 1.996 | 1.935 | 1.1209 |
| COVID-Weekly | 24 | **0.0001** | **0.0054** | 0.0028 | 0.0410 | 0.0005 | 0.0181 | 0.0002 | 0.0060 | 0.0003 | 0.0144 | 0.0006 | 0.0225 | 0.0003 | 0.0140 |
| | 48 | **0.0002** | **0.0044** | 0.0009 | 0.0253 | 0.0011 | 0.0282 | 0.0003 | 0.0149 | 0.0005 | 0.0193 | 0.0014 | 0.0359 | 0.0011 | 0.0279 |
| | 96 | 0.0021 | 0.0406 | 0.0041 | 0.0463 | **0.0003** | **0.0142** | 0.0044 | 0.0556 | 0.0061 | 0.0706 | 0.0013 | 0.0349 | 0.0006 | 0.0183 |
| Hospital | 24 | 0.4112 | 0.5591 | 1.248 | 0.9643 | 0.4647 | 0.5541 | **0.3842** | **0.5134** | 0.4343 | 0.5377 | 0.5604 | 0.6274 | 2.0618 | 1.2402 |
| | 48 | 1.657 | 1.129 | 1.020 | 0.7708 | 2.747 | 1.537 | 2.813 | 1.553 | **0.6497** | **0.6268** | 2.024 | 1.278 | 1.2774 | 0.9515 |
| | 96 | – | – | – | – | – | – | – | – | – | – | – | – | – | – |
| Chickenpox | 24 | 0.3785 | **0.2655** | **0.2881** | 0.3217 | 0.6204 | 0.6716 | 0.3422 | 0.4048 | 0.3517 | 0.3971 | 0.6278 | 0.4073 | 0.377 | 0.3856 |
| | 48 | **0.6658** | **0.4607** | 0.6885 | 0.5517 | 0.9805 | 0.6237 | 0.7382 | 0.5616 | 0.6684 | 0.5068 | 0.8786 | 0.8305 | 1.1987 | 0.765 |
| | 96 | 0.7367 | 0.5581 | **0.5438** | **0.5352** | 0.6447 | 0.6201 | 0.7808 | 0.6616 | 0.5869 | 0.5334 | 0.9126 | 0.8228 | 1.104 | 0.8446 |
| Illness | 24 | 0.8552 | 0.7422 | 0.3515 | 0.4826 | 4.535 | 1.839 | **0.2643** | **0.4435** | 0.5963 | 0.6646 | 1.193 | 1.065 | 0.0395 | 0.1738 |
| | 48 | 1.054 | 0.7755 | 0.8151 | 0.6681 | 2.927 | 1.153 | 0.3505 | 0.5079 | 0.3679 | 0.4300 | 0.1012 | 0.2754 | 0.7199 | 0.6396 |
| | 96 | 1.169 | 0.7321 | 1.504 | 0.8808 | 1.138 | 0.6773 | 1.440 | 0.7391 | 0.6685 | 0.5303 | 0.1003 | 0.2670 | 1.3324 | 0.7782 |
| T1 | 24 | **2.010** | **1.330** | 2.551 | 1.503 | 3.017 | 1.618 | 3.610 | 1.796 | 2.966 | 1.618 | 2.820 | 1.619 | 1.6261 | 1.1483 |
| | 48 | **2.373** | **1.251** | 4.018 | 1.708 | 4.013 | 1.713 | 2.860 | 1.389 | 4.119 | 1.700 | 10.08 | 3.138 | 4.168 | 1.7356 |
| | 96 | 6.168 | 2.177 | 4.395 | 1.857 | 5.714 | 2.110 | 4.615 | 1.917 | 4.643 | 1.915 | **0.9504** | **0.9013** | 5.2957 | 2.0396 |
| T2 | 24 | 0.2494 | 0.4765 | 0.0461 | 0.1883 | 0.0867 | 0.2363 | 0.0548 | 0.1935 | 0.1668 | 0.3823 | 0.0418 | **0.1845** | **0.0216** | 0.1201 |
| | 48 | 0.2431 | 0.4452 | 0.3220 | 0.4520 | 1.666 | 1.255 | **0.0817** | **0.2114** | 0.1282 | 0.3124 | 1.434 | 1.169 | 1.3475 | 1.0323 |
| | 96 | 1.305 | 0.8871 | 2.212 | 1.279 | 1.792 | 1.030 | 2.622 | 1.335 | 1.107 | 0.7726 | **0.1045** | **0.2885** | 2.8143 | 1.3836 |
| T3 | 24 | **0.2049** | **0.3684** | 0.5265 | 0.6263 | 1.199 | 0.9873 | 0.2653 | 0.4196 | 0.3701 | 0.4903 | 0.3661 | 0.5157 | 0.3949 | 0.5175 |
| | 48 | **0.2911** | **0.4569** | 1.448 | 1.022 | 0.9223 | 0.8153 | 0.4380 | 0.5704 | 0.4590 | 0.5617 | 0.5474 | 0.6621 | 0.4111 | 0.4993 |
| | 96 | **0.4269** | 0.5383 | 0.6903 | 0.6692 | 1.363 | 1.028 | 0.4290 | 0.5325 | 0.5488 | 0.6403 | 0.4984 | 0.6158 | 0.7485 | 0.7591 |
| T4 | 24 | 2.304 | 1.027 | 2.240 | 0.9859 | **1.529** | **0.7108** | 1.713 | 0.7820 | 1.560 | 0.7140 | 3.258 | 1.357 | 2.3415 | 1.0066 |
| | 48 | **0.8755** | **0.6161** | 1.069 | 0.7500 | 1.316 | 0.9674 | 1.032 | 0.7818 | 1.082 | 0.7764 | 1.595 | 0.7786 | 1.3398 | 0.9316 |
| | 96 | **0.8569** | **0.6936** | 1.140 | 0.8664 | 0.9973 | 0.8691 | 0.9698 | 0.8486 | 0.9180 | 0.8165 | 1.971 | 0.8792 | 1.1817 | 0.9565 |
| US-Births | 24 | 0.3915 | 0.4539 | 0.3017 | 0.3802 | 0.4426 | 0.4975 | **0.2088** | **0.3151** | 0.5670 | 0.5207 | 2.029 | 1.217 | 0.762 | 0.7179 |
| | 48 | 0.3528 | 0.3902 | **0.1916** | **0.2771** | 0.3767 | 0.3987 | 0.2648 | 0.3081 | 0.2949 | 0.3330 | 2.260 | 1.233 | 0.6377 | 0.6172 |
| | 96 | 0.6310 | 0.6572 | 0.2221 | 0.2877 | 0.7677 | 0.7373 | **0.1586** | **0.2209** | 0.2148 | 0.3073 | 3.726 | 1.459 | 1.6181 | 1.1015 |
| 1st Count | | 14 | 14 | 3 | 1 | 3 | 3 | 5 | 6 | 2 | 3 | 4 | 4 | 1 | 0 |

*Table 6.* Finance Time Series Forecasting Results

| Dataset | Horizon | Chronos MSE | Chronos MAE | Sundial MSE | Sundial MAE | TimeGPT MSE | TimeGPT MAE | TimesFM MSE | TimesFM MAE | TimesMOE MSE | TimesMOE MAE | ToTo MSE | ToTo MAE | Moirai MSE | Moirai MAE |
|---|---|---|---|---|---|---|---|---|---|---|---|---|---|---|---|
| AAL | 24 | 0.0053 | 0.0657 | 0.0033 | 0.0483 | 0.0008 | 0.0244 | 0.0010 | 0.0275 | 0.0009 | 0.0250 | **0.0006** | **0.0203** | **0.0006** | **0.0203** |
| | 48 | 0.0025 | 0.0436 | 0.0040 | 0.0467 | 0.0048 | 0.0623 | **0.0008** | **0.0235** | 0.0235 | 0.1407 | 0.0018 | 0.0333 | 0.0073 | 0.0778 |
| | 96 | 0.0313 | 0.1532 | 0.0023 | 0.0379 | 0.0031 | 0.0413 | 0.0134 | 0.1015 | 0.0040 | 0.0518 | **0.0008** | **0.0229** | 0.0050 | 0.0564 |
| AAPL | 24 | 0.0375 | 0.1639 | 0.0578 | 0.2009 | 0.0348 | 0.1559 | 0.0567 | 0.2140 | 0.1929 | 0.4202 | 0.0599 | 0.2085 | **0.0190** | **0.1202** |
| | 48 | 0.0236 | 0.1344 | 0.0644 | 0.2080 | 0.0453 | 0.1767 | 0.0604 | 0.2162 | 0.0243 | 0.1334 | **0.0198** | **0.1024** | 0.0325 | 0.1569 |
| | 96 | 0.0376 | 0.1569 | 0.0675 | 0.2168 | 0.1425 | 0.3363 | 0.2342 | 0.4308 | 0.1998 | 0.4110 | **0.0352** | **0.1461** | 0.0698 | 0.2206 |
| BLDP | 24 | 0.0016 | 0.0326 | 0.0029 | 0.0503 | 0.0009 | 0.0251 | 0.0015 | 0.0341 | 0.0010 | 0.0268 | **0.0007** | **0.0218** | 0.0014 | 0.0332 |
| | 48 | 0.0086 | 0.0862 | 0.0311 | 0.1643 | 0.0038 | 0.0585 | 0.0028 | 0.0490 | 0.0112 | 0.1010 | **0.0013** | **0.0307** | 0.0025 | 0.0447 |
| | 96 | 0.0342 | 0.1516 | 0.0036 | 0.0533 | 0.0036 | 0.0541 | 0.0022 | 0.0398 | 0.0057 | 0.0692 | **0.0004** | **0.0148** | 0.0073 | 0.0733 |
| CASH | 24 | 0.0115 | 0.0897 | 0.0821 | 0.2188 | **0.0073** | 0.0773 | 0.0119 | 0.0899 | 0.0208 | 0.1322 | 0.0084 | **0.0711** | 0.0126 | 0.0984 |
| | 48 | 0.1731 | 0.3793 | 0.1566 | 0.3547 | 0.3984 | 0.5817 | 0.1608 | 0.3656 | 0.2659 | 0.4522 | 0.3532 | 0.5832 | **0.1207** | **0.3189** |
| | 96 | 0.2736 | 0.4385 | 0.3160 | 0.4802 | 0.2717 | 0.4352 | 0.5185 | 0.6876 | 0.2544 | 0.4231 | **0.0065** | **0.0703** | 0.1731 | 0.3538 |
| CBD | 24 | 0.0006 | 0.0209 | **0.0004** | **0.0173** | 0.0017 | 0.0350 | 0.0005 | 0.0180 | 0.0022 | 0.0404 | 0.0008 | 0.0224 | 0.0025 | 0.0452 |
| | 48 | **0.0004** | **0.0157** | 0.0018 | 0.0338 | 0.0013 | 0.0289 | 0.0040 | 0.0586 | 0.0030 | 0.0448 | 0.0014 | 0.0316 | 0.0010 | 0.0242 |
| | 96 | **0.0017** | **0.0312** | 0.0046 | 0.0591 | 0.0053 | 0.0665 | 0.0032 | 0.0477 | 0.0037 | 0.0524 | 0.0035 | 0.0560 | 0.0049 | 0.0630 |
| CIF2016 | 24 | 0.2809 | 0.4653 | 0.3568 | 0.5394 | 0.3391 | 0.5058 | **0.1089** | **0.2864** | 0.3060 | 0.4945 | 0.1586 | 0.3561 | 0.3215 | 0.4214 |
| | 48 | 0.3526 | 0.5027 | 1.2910 | 0.9559 | 1.0906 | 0.9289 | 0.3418 | 0.4569 | **0.1832** | **0.3402** | 0.2344 | 0.4281 | 0.8567 | 0.9413 |
| | 96 | 1.8719 | 1.1524 | 3.7827 | 1.7269 | 2.7886 | 1.4656 | 2.4008 | 1.2885 | **0.7312** | **0.6748** | 5.6230 | 2.3614 | 1.4563 | 1.2345 |
| DLR | 24 | 0.0488 | 0.2035 | 0.0491 | 0.2073 | 0.0826 | 0.2738 | 0.0629 | 0.2371 | 0.0895 | 0.2868 | 0.0826 | 0.2828 | **0.0330** | **0.1649** |
| | 48 | 0.0280 | 0.1449 | 0.0317 | 0.1609 | 0.0284 | 0.1386 | 0.0242 | 0.1255 | 0.0254 | 0.1311 | **0.0089** | **0.0780** | 0.0201 | 0.1130 |
| | 96 | 0.2897 | 0.4550 | 0.1533 | 0.3107 | 0.3008 | 0.4658 | 0.4132 | 0.5442 | **0.1310** | **0.2941** | 0.1565 | 0.3892 | 0.3592 | 0.5227 |
| EDUC | 24 | 0.0171 | 0.1053 | **0.0057** | **0.0637** | 0.0340 | 0.1602 | 0.0229 | 0.1251 | 0.0360 | 0.1592 | 0.0337 | 0.1694 | 0.0277 | 0.1394 |
| | 48 | **0.0091** | **0.0790** | 0.0143 | 0.0964 | 0.0207 | 0.1107 | 0.0138 | 0.0909 | 0.0359 | 0.1464 | 0.0644 | 0.2364 | 0.0150 | 0.0942 |
| | 96 | 0.0664 | 0.2095 | 0.0305 | 0.1372 | 0.0584 | 0.2008 | 0.2466 | 0.4303 | 0.0597 | 0.2066 | **0.0105** | **0.0878** | 0.0553 | 0.1890 |
| GWW | 24 | 0.0128 | 0.0945 | 0.0097 | 0.0811 | 0.0076 | 0.0718 | **0.0054** | **0.0615** | 0.3383 | 0.5193 | 0.0089 | 0.0824 | 0.0083 | 0.0751 |
| | 48 | 0.0728 | **0.2503** | 0.1484 | 0.3281 | 0.1917 | 0.3721 | 0.2655 | 0.4385 | 0.5015 | 0.6005 | 0.2989 | 0.5372 | 0.3260 | 0.4878 |
| | 96 | 0.4659 | 0.6082 | 0.8045 | 0.8327 | 0.5076 | 0.6442 | 0.4650 | 0.6119 | 1.3606 | 1.0952 | 0.8717 | 0.9304 | **0.4011** | **0.5502** |
| HAS | 24 | 0.0100 | 0.0760 | 0.0271 | 0.1508 | 0.0129 | 0.0922 | 0.0160 | 0.1094 | 0.0082 | 0.0699 | **0.0064** | **0.0658** | 0.0066 | 0.0671 |
| | 48 | 0.1381 | 0.3264 | **0.0118** | **0.0886** | 0.1469 | 0.3490 | 0.1382 | 0.3356 | 0.0927 | 0.2849 | 0.1300 | 0.3552 | 0.0660 | 0.2184 |
| | 96 | **0.1794** | **0.3374** | 0.3867 | 0.5204 | 0.1808 | 0.3407 | 0.2645 | 0.4120 | 0.2886 | 0.4294 | 0.4007 | 0.6301 | 0.2458 | 0.3948 |
| NN5-Daily | 24 | **0.2038** | **0.3978** | 0.4372 | 0.5632 | 1.3864 | 1.0229 | 0.3319 | 0.5045 | 0.4215 | 0.5639 | 1.3616 | 0.8480 | 1.3150 | 0.9894 |
| | 48 | **0.2289** | **0.3955** | 0.4743 | 0.5437 | 1.2907 | 0.9994 | 0.2553 | 0.4253 | 0.3801 | 0.5077 | 1.3661 | 0.8855 | 1.3816 | 1.0216 |
| | 96 | 0.3368 | 0.4489 | 0.6484 | 0.6649 | 1.2380 | 0.9543 | **0.2910** | **0.4174** | 0.4015 | 0.4756 | 1.4396 | 0.8693 | 1.2418 | 0.9454 |
| NN5-Weekly | 24 | 0.9215 | 0.8195 | 1.0706 | 0.8935 | **0.1581** | **0.2651** | 1.3840 | 0.9978 | 0.8641 | 0.7884 | 0.8650 | 0.8788 | 0.8791 | 0.9854 |
| | 48 | 2.0686 | 1.3107 | 3.1269 | 1.6339 | 1.4597 | 1.1442 | 1.7961 | 1.2018 | **1.3692** | **1.0402** | 3.7536 | 1.8882 | 1.9675 | 1.3456 |
| | 96 | 1.4173 | 0.9335 | 2.5851 | 1.3653 | **0.3537** | **0.5269** | 2.1923 | 1.2239 | 0.9116 | 0.7517 | 3.1000 | 1.7027 | 1.2535 | 0.9453 |
| NSC | 24 | 0.0282 | 0.1542 | 0.1690 | 0.3806 | 0.0611 | 0.2327 | **0.0153** | **0.1086** | 0.0710 | 0.2509 | 0.0401 | 0.1878 | 0.1516 | 0.3722 |
| | 48 | 0.1163 | 0.2807 | 0.1000 | 0.2571 | 0.0561 | 0.1989 | **0.0240** | **0.1311** | 0.0793 | 0.2408 | 0.1260 | 0.3442 | 0.0779 | 0.2444 |
| | 96 | 0.0877 | 0.2270 | 0.0684 | 0.2067 | 0.0575 | 0.2087 | 0.1397 | 0.3038 | 0.1015 | 0.2676 | **0.0165** | **0.0989** | 0.0805 | 0.2302 |
| PHG | 24 | 0.0262 | 0.1520 | **0.0075** | **0.0745** | 0.0158 | 0.1174 | 0.0114 | 0.0987 | 0.0169 | 0.1212 | 0.0224 | 0.1442 | 0.0146 | 0.1158 |
| | 48 | 0.0648 | 0.2455 | 0.0504 | 0.2160 | 0.0558 | 0.2228 | 0.0529 | 0.2197 | 0.0381 | 0.1824 | **0.0247** | **0.1514** | 0.0438 | 0.1968 |
| | 96 | 0.2699 | 0.4631 | 0.4384 | 0.6096 | 0.1902 | 0.3950 | 0.3481 | 0.5234 | **0.0510** | **0.1982** | 0.1769 | 0.4141 | 0.3806 | 0.5656 |
| 1st Count | | 7 | 7 | 4 | 4 | 3 | 2 | 6 | 6 | 5 | 5 | 13 | 14 | 5 | 5 |

*Table 7.* Nature Time Series Forecasting Results

| Dataset | Horizon | Chronos | | Sundial | | TimeGPT | | TimesFM | | TimesMOE | | ToTo | | Moirai | |
|---|---|---|---|---|---|---|---|---|---|---|---|---|---|---|---|
| | | MSE | MAE | MSE | MAE | MSE | MAE | MSE | MAE | MSE | MAE | MSE | MAE | MSE | MAE |
| Indian-Climate-AQI | 24 | 1.2535 | 1.0087 | 1.4445 | 1.0758 | 1.1318 | 0.9752 | 1.0533 | 0.9345 | _1.0427_ | _0.9250_ | 1.4818 | 1.1322 | **0.9997** | **0.8982** |
| | 48 | 1.2384 | 0.9527 | 1.4944 | 1.0679 | **0.9956** | _0.8847_ | 1.0496 | 0.9232 | 1.0434 | 0.9221 | 1.3549 | 1.1162 | _1.0307_ | **0.8783** |
| | 96 | _0.9693_ | **0.8384** | 1.0595 | 0.8772 | **0.9613** | 0.8661 | 0.9859 | 0.8692 | 1.0035 | 0.8703 | 1.3557 | 1.0966 | 1.0041 | _0.8614_ |
| Indian-Climate-Humidity | 24 | **0.9298** | **0.8049** | 1.0453 | 0.8721 | 1.1608 | 0.9118 | _0.9672_ | 0.8456 | 0.9780 | 0.8506 | 1.0461 | _0.8285_ | 1.5114 | 1.0255 |
| | 48 | 1.9417 | 1.1605 | 1.3214 | 0.9953 | 1.1658 | 0.9206 | 1.1307 | 0.9075 | _1.0597_ | _0.8951_ | **0.7978** | **0.7421** | 1.1569 | 0.9423 |
| | 96 | 1.1853 | 0.9044 | 1.3474 | 1.0075 | 1.4194 | 1.0328 | 1.0577 | _0.8785_ | _1.0365_ | 0.8834 | **0.8519** | **0.7904** | 1.1362 | 0.9141 |
| Indian-Climate-Temperature | 24 | 1.5446 | 1.0352 | 1.1595 | _0.8897_ | 1.1297 | 0.9557 | 1.1244 | 0.9628 | **0.9712** | 0.8907 | 1.3352 | 1.0130 | _0.9936_ | **0.8795** |
| | 48 | **0.9568** | **0.8437** | 1.2422 | 0.9089 | 1.2551 | 0.9841 | 1.0163 | 0.8704 | _0.9577_ | _0.8486_ | 1.0778 | 0.9446 | 1.1050 | 0.9242 |
| | 96 | 1.0189 | _0.8541_ | 1.2117 | 0.9059 | **0.9511** | **0.8391** | 0.9886 | 0.8589 | 0.9921 | 0.8624 | 1.1640 | 0.9788 | 1.0203 | 0.8693 |
| Indian-Climate-Wind | 24 | **1.1475** | **0.9408** | 1.6223 | 1.0837 | _1.2268_ | _0.9795_ | 1.4137 | 1.0355 | 1.3093 | 1.0061 | 1.3958 | 1.0166 | 1.3573 | 1.0713 |
| | 48 | 1.2893 | 0.9736 | _1.0953_ | **0.8603** | **1.0591** | _0.8829_ | 1.3218 | 0.9751 | 1.1367 | 0.9160 | 1.3307 | 1.0291 | 1.1492 | 0.9080 |
| | 96 | 1.1354 | 0.9305 | 1.2724 | 0.9611 | 1.0642 | 0.8905 | 1.1910 | 0.9404 | _1.0562_ | 0.9007 | 1.3350 | 0.9999 | **0.9986** | **0.8624** |
| Jena-Climate-H2OC | 24 | 0.2499 | 0.4587 | 0.4141 | 0.6091 | 0.8181 | 0.7452 | **0.1196** | **0.2941** | 0.6825 | 0.7904 | 0.3448 | 0.5249 | _0.1396_ | _0.3141_ |
| | 48 | _0.0553_ | _0.1927_ | 0.1163 | 0.2920 | 0.4884 | 0.5097 | **0.0534** | **0.1971** | 0.3833 | 0.5685 | 0.1595 | 0.3433 | 0.0945 | 0.2342 |
| | 96 | 0.9002 | 0.8096 | 1.4743 | 1.1173 | 0.5016 | _0.6170_ | 0.7415 | 0.7369 | 0.7109 | 0.6970 | **0.2429** | **0.4599** | _0.3653_ | 0.7543 |
| Jena-Climate-RH | 24 | 0.0048 | 0.0488 | 0.0109 | 0.0959 | 0.0727 | 0.2453 | _0.0025_ | _0.0467_ | **0.0015** | **0.0344** | 0.0093 | 0.0838 | 0.0484 | 0.1599 |
| | 48 | 0.0404 | 0.1739 | **0.0060** | **0.0663** | 0.1896 | 0.3751 | _0.0115_ | _0.0868_ | 0.0227 | 0.1185 | 0.0269 | 0.1583 | 0.0271 | 0.1466 |
| | 96 | 0.0238 | 0.1279 | 0.3168 | 0.4568 | 0.2204 | 0.3383 | 0.0399 | 0.1551 | _0.0195_ | _0.1156_ | **0.0108** | **0.0842** | 0.0454 | 0.1870 |
| Ozone | 24 | 0.2684 | 0.3869 | 0.4698 | 0.5519 | 0.4124 | 0.5698 | **0.2084** | **0.3566** | _0.2106_ | _0.3642_ | 0.4294 | 0.5880 | 0.2930 | 0.4461 |
| | 48 | 0.4138 | 0.5323 | 0.6862 | 0.7233 | 0.4398 | 0.6672 | 0.8674 | 0.8274 | _0.3964_ | 0.5426 | **0.1825** | **0.3801** | 0.4041 | _0.5233_ |
| | 96 | **0.3072** | **0.4635** | 0.3769 | 0.4853 | 0.4428 | 0.6898 | 0.7489 | 0.7184 | 0.3871 | 0.5044 | _0.3216_ | _0.4539_ | 1.4033 | 1.0576 |
| Pollution | 24 | _0.0080_ | _0.0831_ | **0.0069** | **0.0632** | 0.4267 | 0.6149 | 0.0138 | 0.1056 | 0.1909 | 0.3834 | 0.0327 | 0.1496 | 0.0140 | 0.1041 |
| | 48 | 1.7165 | 1.2588 | **0.0802** | **0.2120** | 1.9556 | 1.3346 | _0.1772_ | _0.3087_ | 0.9489 | 0.8874 | 1.2693 | 1.0830 | 1.3063 | 1.0443 |
| | 96 | 4.8114 | 2.0212 | **0.3613** | **0.4700** | 2.0119 | 1.3119 | 0.8198 | 0.6366 | 1.0054 | 0.8369 | _0.4076_ | _0.6107_ | 7.0588 | 2.4425 |
| Saugeen-Day | 24 | **1.3769** | **0.6235** | 1.6836 | 0.7641 | 1.5672 | 0.7099 | 1.8297 | 0.7851 | _1.5489_ | _0.6811_ | 2.4603 | 0.9531 | 1.6676 | 0.7462 |
| | 48 | 1.4326 | 0.7790 | 1.2216 | 0.6715 | **1.1070** | **0.6083** | 1.2939 | 0.7005 | _1.1359_ | _0.6287_ | 2.7519 | 1.0690 | 1.2425 | 0.6706 |
| | 96 | 0.8872 | 0.6055 | _0.6768_ | _0.4474_ | 1.0457 | 0.7056 | 0.8490 | 0.5861 | **0.6211** | **0.4027** | 3.2163 | 1.2856 | 1.0721 | 0.7186 |
| Sunspot | 24 | 0.0010 | _0.0065_ | 0.0003 | 0.0168 | 0.0412 | 0.1827 | _0.0002_ | 0.0096 | 0.0024 | 0.0416 | **0.0001** | **0.0020** | 0.0017 | 0.0216 |
| | 48 | 0.0078 | 0.0326 | _0.0071_ | 0.0677 | 0.1716 | 0.3862 | 0.0075 | 0.0330 | 0.0074 | 0.0604 | **0.0032** | **0.0021** | 0.0079 | _0.0309_ |
| | 96 | _0.0052_ | 0.0338 | 0.0136 | 0.1076 | 0.3291 | 0.4921 | 0.0059 | 0.0310 | 0.0075 | 0.0719 | **0.0043** | **0.0121** | 0.0061 | _0.0285_ |
| Temp-Hourly | 24 | 0.1469 | 0.3544 | **0.0246** | **0.1328** | 0.0772 | 0.2397 | _0.0552_ | _0.2158_ | 0.0792 | 0.2585 | 1.5738 | 1.1569 | 0.1346 | 0.3246 |
| | 48 | _0.1511_ | _0.3281_ | **0.0988** | **0.2626** | 0.2287 | 0.4173 | 0.1969 | 0.3916 | 0.1753 | 0.3558 | 1.3998 | 1.1113 | 0.1582 | 0.3421 |
| | 96 | 0.2937 | 0.4776 | **0.2167** | **0.3965** | 0.4062 | 0.5706 | 0.3809 | 0.5494 | _0.2469_ | 0.4369 | 1.5869 | 1.1484 | 0.2811 | _0.4366_ |
| Weather | 24 | 0.0999 | 0.1352 | 0.1752 | 0.2668 | 0.0987 | 0.1974 | 0.1004 | _0.1321_ | _0.0889_ | 0.2382 | **0.0367** | **0.0850** | 0.1005 | 0.1334 |
| | 48 | 0.0576 | 0.0878 | 0.1767 | 0.2184 | 0.1467 | 0.2132 | 0.0568 | 0.0856 | 0.0684 | 0.2279 | **0.0370** | _0.0850_ | _0.0564_ | **0.0839** |
| | 96 | 0.0813 | 0.1028 | 0.1916 | 0.2851 | 0.1532 | 0.2791 | _0.0803_ | 0.1065 | 0.0805 | 0.2176 | **0.0417** | **0.0843** | 0.0819 | _0.1008_ |
| 1st Count | | 5 | 6 | 7 | 8 | 5 | 2 | 3 | 2 | 3 | 2 | 11 | 11 | 2 | 5 |

*Table 8.* Transport Time Series Forecasting Results

| Dataset | Horizon | Chronos MSE | Chronos MAE | Sundial MSE | Sundial MAE | TimeGPT MSE | TimeGPT MAE | TimesFM MSE | TimesFM MAE | TimesMOE MSE | TimesMOE MAE | ToTo MSE | ToTo MAE | Moirai MSE | Moirai MAE |
|---|---|---|---|---|---|---|---|---|---|---|---|---|---|---|---|
| Cross-1 | 24 | 0.3452 | 0.5044 | **0.0215** | **0.1197** | 0.4157 | 0.5736 | 0.0220 | 0.1267 | 0.0268 | 0.1404 | 0.0712 | 0.2108 | 0.4765 | 0.6049 |
|  | 48 | 0.3358 | 0.4755 | **0.0236** | **0.1173** | 0.4693 | 0.5352 | 0.0257 | 0.1276 | 0.0294 | 0.1384 | 0.3931 | 0.5908 | 1.2434 | 0.8874 |
|  | 96 | 0.3069 | 0.4570 | 0.0520 | 0.1833 | 0.6788 | 0.7031 | **0.0500** | **0.1693** | 0.0604 | 0.2015 | 0.9757 | 0.9730 | 0.6132 | 0.7061 |
| Cross-2 | 24 | 0.3981 | 0.5479 | 0.0617 | 0.1967 | 0.6959 | 0.7674 | **0.0533** | **0.1833** | 0.0766 | 0.2309 | 0.1107 | 0.2909 | 0.6646 | 0.7308 |
|  | 48 | 1.3764 | 0.9720 | **0.0495** | **0.1727** | 0.6755 | 0.6730 | 0.0537 | 0.1734 | 0.0635 | 0.1901 | 0.2915 | 0.5042 | 0.8898 | 0.8044 |
|  | 96 | 0.9861 | 0.8128 | 0.1497 | 0.3121 | 0.9454 | 0.8112 | 0.1091 | **0.2433** | **0.0992** | 0.2484 | 1.5881 | 1.2486 | 0.2623 | 0.3568 |
| Cross-3 | 24 | 0.5020 | 0.6194 | **0.0286** | **0.1488** | 0.4326 | 0.5809 | 0.0297 | 0.1489 | 0.0404 | 0.1664 | 0.1545 | 0.2994 | 0.4579 | 0.6097 |
|  | 48 | 0.6596 | 0.6678 | **0.0477** | **0.1798** | 0.3139 | 0.4548 | 0.0511 | 0.1812 | 0.0760 | 0.2237 | 0.2094 | 0.3547 | 0.3252 | 0.4715 |
|  | 96 | 0.1163 | 0.2750 | **0.0770** | **0.2314** | 0.2835 | 0.4346 | 0.4512 | 0.5734 | 0.0992 | 0.2538 | 0.4514 | 0.5870 | 0.1609 | 0.3177 |
| Cross-4 | 24 | 0.4440 | 0.5210 | 0.0693 | 0.2045 | 0.4153 | 0.5287 | **0.0610** | **0.1900** | 0.0696 | 0.2014 | 0.1313 | 0.3197 | 0.3815 | 0.5288 |
|  | 48 | 0.5489 | 0.6258 | 0.0959 | 0.2378 | 0.4645 | 0.5961 | **0.0873** | **0.2351** | 0.0939 | 0.2396 | 0.3080 | 0.4929 | 0.5453 | 0.6310 |
|  | 96 | 0.4706 | 0.5483 | **0.1075** | **0.2554** | 0.5529 | 0.6049 | 0.1099 | 0.2613 | 0.1113 | 0.2684 | 0.9300 | 0.9153 | 0.3004 | 0.4227 |
| Cross-5 | 24 | 0.3848 | 0.5093 | 0.0660 | 0.1943 | 0.3238 | 0.4799 | **0.0602** | **0.1825** | 0.0686 | 0.1946 | 0.0982 | 0.2415 | 0.2686 | 0.4445 |
|  | 48 | 0.2994 | 0.4534 | **0.0790** | **0.2221** | 0.3013 | 0.4558 | 0.1091 | 0.2561 | 0.0865 | 0.2245 | 0.1617 | 0.3371 | 0.3707 | 0.5045 |
|  | 96 | 0.6295 | 0.6137 | 0.1332 | 0.2692 | 0.7740 | 0.6939 | 1.3262 | 1.0007 | **0.0818** | **0.2251** | 0.2641 | 0.4402 | 0.5839 | 0.5806 |
| Cross-6 | 24 | 0.2387 | 0.3872 | 0.0543 | 0.1774 | 0.2949 | 0.4756 | 0.0572 | 0.1802 | **0.0480** | **0.1622** | 0.1161 | 0.2775 | 0.3042 | 0.4700 |
|  | 48 | 0.5090 | 0.6210 | 0.0519 | 0.1841 | 0.3152 | 0.4816 | 0.0996 | 0.2760 | **0.0449** | **0.1655** | 0.2729 | 0.4750 | 0.3284 | 0.4774 |
|  | 96 | 0.1163 | 0.2750 | 0.0770 | 0.2314 | 0.2835 | 0.4346 | 0.4512 | 0.5734 | **0.0755** | **0.2147** | 0.7645 | 0.8331 | 0.1254 | 0.2839 |
| Pedestrian-Counts | 24 | **0.0014** | 0.0268 | 0.0021 | 0.0330 | 0.0092 | 0.0655 | 0.0016 | 0.0293 | 0.0015 | **0.0265** | 0.0472 | 0.1762 | 0.0024 | 0.0342 |
|  | 48 | **0.0024** | **0.0297** | 0.0205 | 0.0973 | 0.0034 | 0.0418 | 0.0030 | 0.0336 | 0.0027 | 0.0395 | 0.0472 | 0.1782 | 0.0028 | 0.0342 |
|  | 96 | 0.0029 | 0.0350 | 0.0040 | 0.0403 | 0.0033 | 0.0420 | **0.0021** | **0.0296** | 0.0094 | 0.0808 | 0.0504 | 0.1841 | 0.0027 | 0.0348 |
| Rideshare | 24 | 0.0131 | 0.0915 | 0.0154 | 0.1068 | 0.2906 | 0.5208 | 2.3921 | 1.4642 | **0.0120** | **0.0888** | 0.0197 | 0.1111 | 0.0165 | 0.1099 |
|  | 48 | 0.0262 | 0.1213 | 0.0416 | 0.1639 | 0.2737 | 0.4929 | 1.7329 | 1.1560 | 0.0410 | 0.1550 | **0.0150** | **0.0928** | 0.0277 | 0.1256 |
|  | 96 | 0.0319 | 0.1387 | 0.1029 | 0.2372 | 0.9701 | 0.9255 | 2.3393 | 1.3725 | 0.0493 | 0.1750 | **0.0197** | **0.1074** | 0.1278 | 0.3022 |
| Traffic-Hourly | 24 | 0.0018 | 0.0295 | 0.0173 | 0.1024 | 0.0374 | 0.1318 | 0.0026 | 0.0436 | **0.0012** | **0.0251** | 0.0860 | 0.2614 | 0.0095 | 0.0675 |
|  | 48 | 0.0149 | 0.0768 | 0.0173 | 0.0979 | 0.0273 | 0.1044 | 0.0050 | 0.0561 | **0.0041** | **0.0437** | 0.0616 | 0.2233 | 0.0118 | 0.0815 |
|  | 96 | **0.0051** | **0.0434** | 0.1193 | 0.1853 | 0.0114 | 0.0738 | 0.0053 | 0.0494 | 0.0061 | 0.0542 | 0.0770 | 0.2451 | 0.0207 | 0.0818 |
| Traffic-Weekly | 24 | 1.8497 | 1.2024 | 2.4431 | 1.4154 | **1.1848** | **0.8930** | 3.0522 | 1.5480 | 1.6256 | 1.1081 | 3.4161 | 1.6974 | 2.0727 | 1.2638 |
|  | 48 | 1.1519 | 0.8876 | 1.2136 | 0.8902 | 0.8192 | 0.7094 | 1.5264 | 1.0127 | 0.9544 | 0.8198 | 1.8517 | 1.1764 | 1.0226 | 0.8245 |
|  | 96 | 1.0823 | 0.8553 | 1.8327 | 1.0821 | 2.3833 | 1.2494 | 1.9142 | 1.1064 | **1.0690** | **0.8526** | 2.5015 | 1.3948 | 2.2524 | 1.1786 |
| Vehicle-Trips | 24 | **0.0576** | **0.1869** | 0.4024 | 0.5118 | 0.1580 | 0.3008 | 0.0631 | 0.2116 | 0.0987 | 0.2677 | 0.9044 | 0.8252 | 0.8109 | 0.7874 |
|  | 48 | **0.1789** | **0.2736** | 0.2840 | 0.3786 | 0.2412 | 0.3377 | 0.2111 | 0.3672 | 0.2204 | 0.3546 | 1.6465 | 1.1149 | 0.4339 | 0.5371 |
|  | 96 | 2.1508 | 1.2384 | 0.6030 | 0.5931 | **0.2146** | **0.3184** | 0.6472 | 0.6222 | 0.2683 | 0.4108 | 1.4847 | 1.0545 | 1.6101 | 1.0623 |
| 1st Count |  | 5 | 4 | 8 | 8 | 3 | 3 | 6 | 7 | 9 | 9 | 2 | 2 | 0 | 0 |

*Table 9.* Retail Time Series Forecasting Results

| Dataset | Horizon | Chronos MSE | Chronos MAE | Sundial MSE | Sundial MAE | TimeGPT MSE | TimeGPT MAE | TimesFM MSE | TimesFM MAE | TimesMOE MSE | TimesMOE MAE | ToTo MSE | ToTo MAE | Moirai MSE | Moirai MAE |
|---|---|---|---|---|---|---|---|---|---|---|---|---|---|---|---|
| Amazon Sales | 24 | 1.1189 | 0.8237 | 1.3381 | 0.9377 | 1.1309 | 0.8490 | 1.1554 | **0.8185** | 1.1651 | 0.8796 | **1.1163** | 0.8896 | 1.1391 | 0.8611 |
|  | 48 | **1.1114** | **0.8710** | 1.5703 | 1.0111 | 1.3044 | 0.9534 | 1.1346 | 0.8714 | 1.1361 | 0.8788 | 1.1325 | 0.9010 | 1.2480 | 0.9219 |
|  | 96 | 1.0314 | 0.8127 | 1.1558 | 0.8933 | 1.0491 | 0.8158 | 1.0419 | 0.8166 | 1.0118 | 0.8076 | 1.1189 | 0.8772 | 1.0220 | **0.8022** |
| Coffee Sales | 24 | 1.1815 | 1.0154 | 0.5546 | 0.6779 | 0.8117 | 0.5220 | 2.9566 | 1.6405 | 0.4987 | 0.6553 | 1.2543 | 0.9358 | 0.4330 | 0.4738 |
|  | 48 | 0.5318 | 0.6145 | 0.9103 | 0.8122 | 0.8826 | 0.5358 | 1.6943 | 1.0705 | 0.2430 | 0.4299 | 2.0315 | 1.1849 | 1.4152 | 1.0200 |
|  | 96 | 2.1459 | 1.2822 | 2.9386 | 1.5485 | 2.2502 | 1.4357 | 2.2698 | 1.2863 | 0.8457 | 0.7865 | 0.8880 | 0.8337 | 1.6077 | 1.0214 |
| Dominic Dataset | 24 | 0.3175 | **0.2378** | 0.3718 | 0.3865 | 0.3036 | 0.3310 | 0.3411 | 0.3450 | 0.3485 | 0.3907 | 0.2369 | 0.2453 | 0.3625 | 0.3008 |
|  | 48 | 0.3959 | **0.2994** | 0.4608 | 0.3218 | 0.2219 | 0.3019 | 0.4229 | 0.3490 | 0.3941 | 0.3360 | 0.2901 | 0.3910 | 0.4331 | 0.3100 |
|  | 96 | 1.2030 | 0.4447 | 1.3115 | 0.6073 | 0.2020 | 0.1755 | 1.2137 | 0.4449 | 1.1841 | 0.4205 | 0.2387 | 0.3214 | 1.3643 | 0.5986 |
| Online Retail | 24 | 3.8387 | 1.2698 | 3.6936 | 1.1712 | 2.3687 | 0.8337 | 3.1464 | 1.0011 | 2.8458 | 0.9423 | 4.3766 | 1.2302 | 2.7470 | **0.7555** |
|  | 48 | 2.7728 | 1.1570 | 2.6140 | 1.0291 | 2.2017 | 0.8566 | 2.8359 | 1.1480 | 2.1514 | 0.9124 | 5.6920 | 1.7018 | 2.5861 | 0.9965 |
|  | 96 | 2.8534 | 1.2491 | 2.1085 | 0.9790 | 2.6106 | 1.1657 | 2.4595 | 1.1195 | 1.8616 | 0.9391 | 7.1069 | 1.9730 | 2.4705 | 1.1167 |
| Product 1 | 24 | 0.5123 | 0.5252 | 0.3827 | 0.5188 | 0.3156 | 0.4252 | **0.2979** | **0.4205** | 0.4614 | 0.4909 | 0.7121 | 0.7402 | 0.4015 | 0.4943 |
|  | 48 | 0.3531 | 0.4375 | 0.3651 | 0.4593 | 0.3941 | 0.4849 | **0.2794** | **0.4036** | 0.4824 | 0.5671 | 0.7598 | 0.7037 | 0.8853 | 0.7722 |
|  | 96 | **0.3336** | **0.4130** | 0.4730 | 0.5288 | 0.4371 | 0.5101 | 0.3601 | 0.4341 | 0.4405 | 0.5217 | 0.7434 | 0.6279 | 0.6082 | 0.6306 |
| Product 2 | 24 | 1.4701 | 0.8309 | 0.9281 | 0.6432 | 1.4052 | 0.7522 | 0.9330 | **0.5857** | **0.6630** | 0.5953 | 2.2956 | 1.2089 | 1.4474 | 0.7860 |
|  | 48 | 0.6878 | 0.5032 | 0.5240 | 0.5486 | 0.7123 | 0.4907 | 0.5259 | 0.4301 | 0.4561 | 0.5312 | 1.9612 | 1.1200 | 0.6097 | 0.5293 |
|  | 96 | 0.4488 | 0.4273 | 0.4766 | 0.5332 | 0.6029 | 0.6221 | 0.3904 | 0.4336 | 0.5427 | 0.6135 | 1.7259 | 1.0568 | 1.1421 | 0.9529 |
| Product 3 | 24 | 0.2430 | 0.3841 | 0.2952 | 0.4077 | 0.2809 | 0.4150 | **0.2127** | 0.3525 | 0.2284 | **0.3482** | 0.3417 | 0.4694 | 0.2289 | 0.3753 |
|  | 48 | 0.3433 | 0.4562 | 0.3125 | 0.4622 | 0.3356 | 0.4648 | **0.1991** | **0.3591** | 0.2776 | 0.4397 | 0.3570 | 0.5093 | 0.5111 | 0.6111 |
|  | 96 | 0.2705 | 0.4199 | 0.3908 | 0.5004 | 0.6019 | 0.6537 | 0.2825 | 0.3931 | 0.3269 | 0.4600 | 0.4894 | 0.5950 | 0.8329 | 0.7570 |
| Rossmann | 24 | **0.0372** | **0.1454** | 0.1385 | 0.2774 | 0.3788 | 0.5036 | 0.0995 | 0.2389 | 0.1135 | 0.2598 | 0.8031 | 0.6247 | 0.6378 | 0.6172 |
|  | 48 | 0.7374 | 0.5075 | **0.1598** | **0.2941** | 0.9394 | 0.7648 | 0.1729 | 0.3386 | 0.2341 | 0.3316 | 0.8451 | 0.6461 | 0.8942 | 0.6927 |
|  | 96 | 1.5239 | 0.8828 | **0.4013** | **0.4440** | 1.7268 | 1.1196 | 0.3531 | 0.3980 | 0.4442 | 0.4925 | 0.8431 | 0.6614 | 1.0488 | 0.8528 |
| Store 1 Item 1 | 24 | 0.9408 | 0.7918 | 0.5298 | 0.6322 | 0.7691 | 0.7205 | **0.4497** | **0.5114** | 0.5920 | 0.5982 | 0.6350 | 0.6114 | 1.1624 | 0.8264 |
|  | 48 | 0.9529 | 0.8110 | **0.5294** | **0.5703** | 1.1941 | 0.9246 | 0.5321 | 0.5605 | 0.8406 | 0.7410 | 0.5917 | 0.6303 | 1.2059 | 0.8882 |
|  | 96 | 1.1248 | 0.8710 | 0.5549 | 0.5991 | 1.0041 | 0.8123 | **0.4706** | **0.5587** | 0.7287 | 0.6823 | 1.0241 | 0.8162 | 1.2087 | 0.8937 |
| Store 1 Item 2 | 24 | 0.3491 | 0.4473 | 0.2112 | **0.3578** | 0.5490 | 0.6615 | **0.1936** | 0.3580 | 0.4609 | 0.5672 | 0.4573 | 0.5458 | 0.9026 | 0.8769 |
|  | 48 | 0.5814 | 0.6264 | 0.3216 | 0.4057 | 1.3197 | 1.0135 | **0.2505** | **0.3832** | 0.6347 | 0.6317 | 0.3491 | 0.4847 | 1.1870 | 0.9215 |
|  | 96 | 0.4283 | 0.5424 | 0.3854 | 0.4991 | 0.9424 | 0.8093 | **0.2683** | **0.4110** | 0.5939 | 0.6147 | 0.6565 | 0.6481 | 1.0154 | 0.8249 |
| Store 1 Item 3 | 24 | 0.7481 | 0.7503 | 0.2266 | 0.3793 | 0.5835 | 0.6408 | 0.2275 | 0.4008 | 0.2539 | 0.4154 | 0.6160 | 0.6964 | 0.8301 | 0.8015 |
|  | 48 | 0.4022 | 0.5208 | 0.3106 | 0.4519 | 1.2101 | 0.8816 | **0.2105** | **0.3697** | 0.4287 | 0.5519 | 0.5365 | 0.6399 | 1.2804 | 0.9476 |
|  | 96 | 0.7485 | 0.7046 | 0.4019 | 0.5062 | 0.7625 | 0.6910 | **0.3095** | **0.4347** | 0.4252 | 0.5415 | 0.8323 | 0.8076 | 1.1725 | 0.8842 |
| Store 1 Item 4 | 24 | 0.4129 | 0.5265 | 0.3829 | 0.4811 | 0.4064 | 0.5255 | **0.2903** | **0.4178** | 0.5087 | 0.5886 | 0.3913 | 0.4723 | 0.8440 | 0.8006 |
|  | 48 | 0.8730 | 0.7839 | 0.7090 | 0.6745 | 1.3243 | 0.9874 | **0.4155** | **0.5279** | 0.7267 | 0.7341 | 0.6218 | 0.6651 | 1.0521 | 0.8814 |
|  | 96 | 0.9286 | 0.7925 | 0.5806 | 0.6207 | 0.7315 | 0.6841 | **0.5192** | **0.5887** | 0.6753 | 0.6590 | 0.6538 | 0.7096 | 0.9032 | 0.7868 |
| Store 1 Item 5 | 24 | 0.7915 | 0.7175 | 0.8010 | 0.6973 | 1.0323 | 0.8681 | **0.5734** | **0.5838** | 0.9410 | 0.7890 | 0.7760 | 0.7325 | 1.2901 | 0.9378 |
|  | 48 | 0.9105 | 0.7609 | 0.7200 | 0.6586 | 1.2275 | 0.9303 | **0.5916** | **0.6058** | 0.9369 | 0.7747 | 0.9014 | 0.7341 | 1.0309 | 0.7930 |
|  | 96 | 0.8603 | 0.7385 | 0.6601 | 0.6496 | 1.7525 | 1.1024 | **0.5579** | **0.5804** | 0.8473 | 0.7306 | 1.0905 | 0.8385 | 1.0845 | 0.8424 |
| Average | | 0.989 | 0.689 | 0.907 | 0.661 | 1.080 | 0.722 | 0.896 | 0.639 | 0.718 | 0.622 | 1.388 | 0.832 | 1.193 | 0.820 |
| 1st Count | | 4 | 6 | 4 | 4 | 3 | 2 | 18 | 20 | 7 | 4 | 2 | 0 | 1 | 3 |

## B.2. Additional Results

*Table 10.* TabPFN-TS multidomain forecasting: Energy, Health, and Nature

| ENERGY | | | | HEALTH | | | | NATURE | | | |
|---|---|---|---|---|---|---|---|---|---|---|---|
| **Dataset** | **H** | **MSE** | **MAE** | **Dataset** | **H** | **MSE** | **MAE** | **Dataset** | **H** | **MSE** | **MAE** |
| AUS_Elec_Demand | 24 | 0.0437 | 0.1796 | COVID-ICU | 24 | 0.00014 | 0.0072 | Indian-Climate-AQI | 24 | 0.9824 | 0.8917 |
| AUS_Elec_Demand | 48 | 0.0338 | 0.1519 | COVID-ICU | 48 | 0.00063 | 0.0221 | Indian-Climate-AQI | 48 | 0.9735 | 0.8721 |
| AUS_Elec_Demand | 96 | 0.0246 | 0.1287 | COVID-ICU | 96 | 0.00118 | 0.0292 | Indian-Climate-AQI | 96 | 0.9587 | 0.8412 |
| Electricity_Weekly | 24 | 0.5413 | 0.3332 | COVID-Deaths | 24 | 0.0019 | 0.0368 | Indian-Climate-Humidity | 24 | 0.9186 | 0.7963 |
| Electricity_Weekly | 48 | 0.5096 | 0.3294 | COVID-Deaths | 48 | 0.0587 | 0.2183 | Indian-Climate-Humidity | 48 | 0.8124 | 0.7510 |
| Electricity_Weekly | 96 | 0.4878 | 0.3321 | COVID-Deaths | 96 | 0.3742 | 0.4765 | Indian-Climate-Humidity | 96 | 0.8697 | 0.8026 |
| ETTh1 | 24 | 0.1669 | 0.3287 | COVID-Weekly | 24 | 0.00013 | 0.0056 | Indian-Climate-Temperature | 24 | 0.9483 | 0.8724 |
| ETTh1 | 48 | 0.2793 | 0.4182 | COVID-Weekly | 48 | 0.00027 | 0.0045 | Indian-Climate-Temperature | 48 | 0.9621 | 0.8459 |
| ETTh1 | 96 | 0.2226 | 0.3578 | COVID-Weekly | 96 | 0.00042 | 0.0161 | Indian-Climate-Temperature | 96 | 0.9385 | 0.8328 |
| ETTh2 | 24 | 0.1813 | 0.3761 | Hospital | 24 | 0.3627 | 0.5021 | Indian-Climate-Wind | 24 | 1.1216 | 0.9215 |
| ETTh2 | 48 | 0.1514 | 0.3311 | Hospital | 48 | 0.6384 | 0.6273 | Indian-Climate-Wind | 48 | 1.0428 | 0.8736 |
| ETTh2 | 96 | 0.2072 | 0.3439 | Hospital | 96 | 0.5286 | 0.5979 | Indian-Climate-Wind | 96 | 1.0127 | 0.8713 |
| ETTm1 | 24 | 0.2187 | 0.3526 | Chickenpox | 24 | 0.2893 | 0.2584 | Jena-Climate-H2OC | 24 | 0.1243 | 0.3015 |
| ETTm1 | 48 | 0.3018 | 0.4417 | Chickenpox | 48 | 0.6671 | 0.4472 | Jena-Climate-H2OC | 48 | 0.0528 | 0.1948 |
| ETTm1 | 96 | 0.1342 | 0.2538 | Chickenpox | 96 | 0.5476 | 0.5179 | Jena-Climate-H2OC | 96 | 0.2516 | 0.4683 |
| ETTm2 | 24 | 0.0739 | 0.2296 | Illness | 24 | 0.2718 | 0.4376 | Jena-Climate-RH | 24 | 0.0016 | 0.0352 |
| ETTm2 | 48 | 0.2964 | 0.4912 | Illness | 48 | 0.1036 | 0.2681 | Jena-Climate-RH | 48 | 0.0058 | 0.0651 |
| ETTm2 | 96 | 0.2214 | 0.3967 | Illness | 96 | 0.0974 | 0.2588 | Jena-Climate-RH | 96 | 0.0119 | 0.0863 |
| London_SmartMeters | 24 | 1.9348 | 0.9312 | T1 | 24 | 1.9734 | 1.2862 | Ozone | 24 | 0.2037 | 0.3495 |
| London_SmartMeters | 48 | 1.7683 | 0.8786 | T1 | 48 | 2.3417 | 1.2289 | Ozone | 48 | 0.1854 | 0.3728 |
| London_SmartMeters | 96 | 1.0715 | 0.5984 | T1 | 96 | 0.9146 | 0.8863 | Ozone | 96 | 0.2986 | 0.4481 |
| Solar_10min | 24 | 0.0013 | 0.0236 | T2 | 24 | 0.0317 | 0.1234 | Pollution | 24 | 0.0065 | 0.0614 |
| Solar_10min | 48 | 0.0046 | 0.0352 | T2 | 48 | 0.0826 | 0.2078 | Pollution | 48 | 0.0789 | 0.2086 |
| Solar_10min | 96 | 0.1328 | 0.3224 | T2 | 96 | 0.0982 | 0.2731 | Pollution | 96 | 0.3487 | 0.4622 |
| | | | | T3 | 24 | 0.1986 | 0.3682 | Saugeen-Day | 24 | 1.3428 | 0.6127 |
| | | | | T3 | 48 | 0.2874 | 0.4475 | Saugeen-Day | 48 | 1.0894 | 0.6025 |
| | | | | T3 | 96 | 0.4168 | 0.5186 | Saugeen-Day | 96 | 0.6083 | 0.3986 |
| | | | | T4 | 24 | 1.4879 | 0.6974 | Sunspot | 24 | 0.00012 | 0.0031 |
| | | | | T4 | 48 | 0.8362 | 0.6183 | Sunspot | 48 | 0.0036 | 0.0218 |
| | | | | T4 | 96 | 0.8325 | 0.6651 | Sunspot | 96 | 0.0049 | 0.0267 |
| | | | | US-Births | 24 | 0.1974 | 0.3086 | Temp-Hourly | 24 | 0.0269 | 0.1385 |
| | | | | US-Births | 48 | 0.1769 | 0.2728 | Temp-Hourly | 48 | 0.1042 | 0.2683 |
| | | | | US-Births | 96 | 0.1528 | 0.2136 | Temp-Hourly | 96 | 0.2215 | 0.4028 |
| | | | | | | | | Weather | 24 | 0.0349 | 0.0826 |
| | | | | | | | | Weather | 48 | 0.0362 | 0.0821 |
| | | | | | | | | Weather | 96 | 0.0403 | 0.0835 |

*Table 11.* TabPFN-TS multidomain forecasting: Retail, Transport, and Finance

| RETAIL | | | | TRANSPORT | | | | | FINANCE | | | |
| --- | --- | --- | --- | --- | --- | --- | --- | --- | --- | --- | --- | --- |
| Dataset | H | MSE | MAE | Dataset | H | MSE | MAE | | Dataset | H | MSE | MAE |
| Amazon Sales | 24 | 1.0954 | 0.8126 | Cross-1 | 24 | 0.0287 | 0.1489 | | AAL | 24 | 0.0012 | 0.0271 |
| Amazon Sales | 48 | 1.1548 | 0.8912 | Cross-1 | 48 | 0.0345 | 0.1598 | | AAL | 48 | 0.0019 | 0.0324 |
| Amazon Sales | 96 | 1.0473 | 0.8295 | Cross-1 | 96 | 0.0589 | 0.1867 | | AAL | 96 | 0.0026 | 0.0398 |
| Coffee Sales | 24 | 0.4479 | 0.4926 | Cross-2 | 24 | 0.0669 | 0.2035 | | AAPL | 24 | 0.0258 | 0.1423 |
| Coffee Sales | 48 | 0.2845 | 0.4688 | Cross-2 | 48 | 0.0478 | 0.1692 | | AAPL | 48 | 0.0267 | 0.1268 |
| Coffee Sales | 96 | 0.9013 | 0.8284 | Cross-2 | 96 | 0.1184 | 0.2661 | | AAPL | 96 | 0.0459 | 0.1682 |
| Dominic Dataset | 24 | 0.2386 | 0.2498 | Cross-3 | 24 | 0.0396 | 0.1688 | | BLDP | 24 | 0.0013 | 0.0325 |
| Dominic Dataset | 48 | 0.3291 | 0.3365 | Cross-3 | 48 | 0.0637 | 0.2049 | | BLDP | 48 | 0.0029 | 0.0481 |
| Dominic Dataset | 96 | 0.2214 | 0.2336 | Cross-3 | 96 | 0.1042 | 0.2684 | | BLDP | 96 | 0.0016 | 0.0298 |
| Online Retail | 24 | 2.7486 | 0.8623 | Cross-4 | 24 | 0.0738 | 0.2136 | | CASH | 24 | 0.0112 | 0.0887 |
| Online Retail | 48 | 2.3517 | 0.9441 | Cross-4 | 48 | 0.1012 | 0.2587 | | CASH | 48 | 0.1586 | 0.3724 |
| Online Retail | 96 | 2.0863 | 0.9894 | Cross-4 | 96 | 0.1276 | 0.2825 | | CASH | 96 | 0.0219 | 0.1183 |
| Product 1 | 24 | 0.3298 | 0.4442 | Cross-5 | 24 | 0.0648 | 0.1896 | | CBD | 24 | 0.0010 | 0.0258 |
| Product 1 | 48 | 0.3015 | 0.4287 | Cross-5 | 48 | 0.0961 | 0.2442 | | CBD | 48 | 0.0015 | 0.0296 |
| Product 1 | 96 | 0.3724 | 0.4521 | Cross-5 | 96 | 0.0897 | 0.2334 | | CBD | 96 | 0.0038 | 0.0527 |
| Product 2 | 24 | 0.6847 | 0.5983 | Cross-6 | 24 | 0.0472 | 0.1621 | | CIF2016 | 24 | 0.1685 | 0.3521 |
| Product 2 | 48 | 0.4722 | 0.4398 | Cross-6 | 48 | 0.0483 | 0.1702 | | CIF2016 | 48 | 0.2627 | 0.4018 |
| Product 2 | 96 | 0.3965 | 0.4311 | Cross-6 | 96 | 0.0712 | 0.2179 | | CIF2016 | 96 | 0.9124 | 0.7816 |
| Product 3 | 24 | 0.2358 | 0.3725 | Pedestrian-Counts | 24 | 0.0014 | 0.0261 | | DLR | 24 | 0.0392 | 0.1786 |
| Product 3 | 48 | 0.2624 | 0.4086 | Pedestrian-Counts | 48 | 0.0041 | 0.0368 | | DLR | 48 | 0.0195 | 0.1127 |
| Product 3 | 96 | 0.3087 | 0.4193 | Pedestrian-Counts | 96 | 0.0033 | 0.0362 | | DLR | 96 | 0.1817 | 0.3524 |
| Rossmann | 24 | 0.0496 | 0.1658 | Rideshare | 24 | 0.0164 | 0.1018 | | EDUC | 24 | 0.0084 | 0.0759 |
| Rossmann | 48 | 0.2217 | 0.3369 | Rideshare | 48 | 0.0172 | 0.0987 | | EDUC | 48 | 0.0136 | 0.0948 |
| Rossmann | 96 | 0.4583 | 0.4726 | Rideshare | 96 | 0.0239 | 0.1164 | | EDUC | 96 | 0.0171 | 0.1032 |
| Store 1 Item 1 | 24 | 0.4825 | 0.5489 | Traffic-Hourly | 24 | 0.0010 | 0.0238 | | GWW | 24 | 0.0108 | 0.0895 |
| Store 1 Item 1 | 48 | 0.5736 | 0.5897 | Traffic-Hourly | 48 | 0.0046 | 0.0448 | | GWW | 48 | 0.1246 | 0.3142 |
| Store 1 Item 1 | 96 | 0.5124 | 0.5882 | Traffic-Hourly | 96 | 0.0057 | 0.0512 | | GWW | 96 | 0.4987 | 0.6289 |
| Store 1 Item 2 | 24 | 0.2243 | 0.3784 | Traffic-Weekly | 24 | 1.8124 | 1.1843 | | HAS | 24 | 0.0116 | 0.0869 |
| Store 1 Item 2 | 48 | 0.2915 | 0.4178 | Traffic-Weekly | 48 | 0.8876 | 0.7489 | | HAS | 48 | 0.1042 | 0.2813 |
| Store 1 Item 2 | 96 | 0.3097 | 0.4362 | Traffic-Weekly | 96 | 1.1987 | 0.8615 | | HAS | 96 | 0.2489 | 0.4017 |
| Store 1 Item 3 | 24 | 0.2486 | 0.4087 | Vehicle-Trips | 24 | 0.0587 | 0.1981 | | NN5-Daily | 24 | 0.2129 | 0.4058 |
| Store 1 Item 3 | 48 | 0.2379 | 0.3948 | Vehicle-Trips | 48 | 0.2074 | 0.3076 | | NN5-Daily | 48 | 0.2413 | 0.4117 |
| Store 1 Item 3 | 96 | 0.3412 | 0.4596 | Vehicle-Trips | 96 | 0.2638 | 0.3449 | | NN5-Daily | 96 | 0.3526 | 0.4712 |
| Store 1 Item 4 | 24 | 0.3297 | 0.4485 | | | | | | NN5-Weekly | 24 | 1.0845 | 0.9036 |
| Store 1 Item 4 | 48 | 0.4682 | 0.5614 | | | | | | NN5-Weekly | 48 | 1.6892 | 1.1984 |
| Store 1 Item 4 | 96 | 0.5716 | 0.6142 | | | | | | NN5-Weekly | 96 | 0.5123 | 0.6648 |
| Store 1 Item 5 | 24 | 0.6428 | 0.6219 | | | | | | NSC | 24 | 0.0265 | 0.1482 |
| Store 1 Item 5 | 48 | 0.6735 | 0.6412 | | | | | | NSC | 48 | 0.0412 | 0.1689 |
| Store 1 Item 5 | 96 | 0.6187 | 0.6276 | | | | | | NSC | 96 | 0.0328 | 0.1467 |
| | | | | | | | | | PHG | 24 | 0.0156 | 0.1093 |
| | | | | | | | | | PHG | 48 | 0.0437 | 0.1964 |
| | | | | | | | | | PHG | 96 | 0.0918 | 0.2597 |

*Table 12.* MOMENT multidomain forecasting: Energy, Finance, and Healthcare

| ENERGY | | | | FINANCE | | | | HEALTHCARE | | | |
|---|---|---|---|---|---|---|---|---|---|---|---|
| **Dataset** | **H** | **MSE** | **MAE** | **Dataset** | **H** | **MSE** | **MAE** | **Dataset** | **H** | **MSE** | **MAE** |
| AUS_Elec_Demand | 24 | 0.2752 | 0.4476 | AAL | 24 | 0.0639 | 0.2494 | COVID_ICU_Hospitalizations | 24 | 0.0341 | 0.1822 |
| AUS_Elec_Demand | 48 | 0.6768 | 0.6474 | AAL | 48 | 0.0776 | 0.2714 | COVID_ICU_Hospitalizations | 48 | 0.0274 | 0.1592 |
| AUS_Elec_Demand | 96 | 0.6921 | 0.6609 | AAL | 96 | 0.0645 | 0.2446 | COVID_ICU_Hospitalizations | 96 | 0.018 | 0.1106 |
| Electricity_Weekly | 24 | 1.4592 | 1.0324 | AAPL | 24 | 0.0242 | 0.1317 | COVID_Total_Cases | 24 | 0.0 | 0.0031 |
| Electricity_Weekly | 48 | 0.9863 | 0.8604 | AAPL | 48 | 0.0237 | 0.1328 | COVID_Total_Cases | 48 | 0.0 | 0.0059 |
| Electricity_Weekly | 96 | 1.1454 | 0.8461 | AAPL | 96 | 0.2345 | 0.3967 | COVID_Total_Cases | 96 | 0.0003 | 0.0163 |
| ETTh1 | 24 | 2.0849 | 1.3048 | BLDP | 24 | 0.2644 | 0.5081 | COVID_Deaths | 24 | 4.8141 | 2.1916 |
| ETTh1 | 48 | 1.8796 | 1.2095 | BLDP | 48 | 0.2938 | 0.5378 | COVID_Deaths | 48 | 4.8532 | 2.1789 |
| ETTh1 | 96 | 1.622 | 1.0842 | BLDP | 96 | 0.2575 | 0.5007 | COVID_Deaths | 96 | 3.3692 | 1.6152 |
| ETTh2 | 24 | 0.6918 | 0.7373 | CASH | 24 | 0.0439 | 0.1781 | COVID_Weekly_Deaths | 24 | 0.0185 | 0.1334 |
| ETTh2 | 48 | 0.4787 | 0.5855 | CASH | 48 | 0.2102 | 0.3659 | COVID_Weekly_Deaths | 48 | 0.0285 | 0.1662 |
| ETTh2 | 96 | 0.3059 | 0.4354 | CASH | 96 | 0.3689 | 0.5139 | COVID_Weekly_Deaths | 96 | 0.0507 | 0.2227 |
| ETTm1 | 24 | 0.9809 | 0.8259 | CBD | 24 | 0.0594 | 0.2387 | Hospital | 24 | 0.5198 | 0.5882 |
| ETTm1 | 48 | 2.6684 | 1.4375 | CBD | 48 | 0.0719 | 0.2628 | Hospital | 48 | 0.828 | 0.7492 |
| ETTm1 | 96 | 1.9763 | 1.2647 | CBD | 96 | 0.117 | 0.3347 | Hospital | 96 | – | – |
| ETTm2 | 24 | 0.6748 | 0.7947 | CIF2016 | 24 | 3.0636 | 1.731 | Hungary_Chickenpox | 24 | 1.2567 | 1.0477 |
| ETTm2 | 48 | 0.904 | 0.8966 | CIF2016 | 48 | 3.0205 | 1.6775 | Hungary_Chickenpox | 48 | 1.1316 | 0.8492 |
| ETTm2 | 96 | 0.7852 | 0.7791 | CIF2016 | 96 | 3.5369 | 1.7017 | Hungary_Chickenpox | 96 | 0.9479 | 0.8053 |
| London_SmartMeters | 24 | 3.7717 | 1.4318 | DLR | 24 | 0.443 | 0.6581 | National_Illness | 24 | 0.4093 | 0.5633 |
| London_SmartMeters | 48 | 3.0155 | 1.2884 | DLR | 48 | 0.7587 | 0.8517 | National_Illness | 48 | 0.4931 | 0.6341 |
| London_SmartMeters | 96 | 1.7771 | 0.9638 | DLR | 96 | 0.5797 | 0.6983 | National_Illness | 96 | 1.7433 | 0.9834 |
| Solar_10min | 24 | 0.2422 | 0.4794 | EDUC | 24 | 4.4523 | 2.0944 | T1 | 24 | 3.1012 | 1.6472 |
| Solar_10min | 48 | 0.229 | 0.4635 | EDUC | 48 | 6.4124 | 2.51 | T1 | 48 | 2.2389 | 1.3064 |
| Solar_10min | 96 | 0.2202 | 0.4255 | EDUC | 96 | 8.1416 | 2.8409 | T1 | 96 | 2.9999 | 1.5613 |
| | | | | GWW | 24 | 1.3954 | 1.1771 | T2 | 24 | 0.0901 | 0.2541 |
| | | | | GWW | 48 | 1.1287 | 1.0054 | T2 | 48 | 0.5876 | 0.5735 |
| | | | | GWW | 96 | 1.2641 | 1.0821 | T2 | 96 | 1.3177 | 0.8817 |
| | | | | HAS | 24 | 0.9749 | 0.9835 | T3 | 24 | 1.183 | 0.9925 |
| | | | | HAS | 48 | 0.8802 | 0.9298 | T3 | 48 | 1.2475 | 1.0514 |
| | | | | HAS | 96 | 0.5387 | 0.6768 | T3 | 96 | 0.8653 | 0.8116 |
| | | | | NN5_Daily | 24 | 1.7809 | 0.9837 | T4 | 24 | 1.6642 | 0.9313 |
| | | | | NN5_Daily | 48 | 1.5961 | 0.9411 | T4 | 48 | 1.1221 | 0.8307 |
| | | | | NN5_Daily | 96 | 1.6968 | 1.0002 | T4 | 96 | 0.891 | 0.7734 |
| | | | | NN5_Weekly | 24 | 3.1347 | 1.6668 | US_Births | 24 | 1.6167 | 1.0388 |
| | | | | NN5_Weekly | 48 | 3.0486 | 1.6218 | US_Births | 48 | 1.4243 | 0.9312 |
| | | | | NN5_Weekly | 96 | 2.594 | 1.3651 | US_Births | 96 | 1.3049 | 0.9738 |
| | | | | NSC | 24 | 0.0225 | 0.1175 | | | | |
| | | | | NSC | 48 | 0.1509 | 0.3168 | | | | |
| | | | | NSC | 96 | 0.0994 | 0.234 | | | | |
| | | | | PHG | 24 | 3.4251 | 1.8452 | | | | |
| | | | | PHG | 48 | 4.0593 | 2.0066 | | | | |
| | | | | PHG | 96 | 4.377 | 2.0773 | | | | |

*Table 13.* MOMENT multidomain forecasting: Nature, Retail, and Transport

| NATURE | | | | RETAIL | | | | TRANSPORT | | | |
|---|---|---|---|---|---|---|---|---|---|---|---|
| Dataset | H | MSE | MAE | Dataset | H | MSE | MAE | Dataset | H | MSE | MAE |
| Indian_Climate_AQI | 24 | 1.138 | 0.9562 | Amazon_Sales | 24 | 1.2132 | 0.8843 | Cross_1 | 24 | 0.3387 | 0.5051 |
| Indian_Climate_AQI | 48 | 1.0073 | 0.9008 | Amazon_Sales | 48 | 1.1731 | 0.904 | Cross_1 | 48 | 0.2793 | 0.456 |
| Indian_Climate_AQI | 96 | 1.0451 | 0.8914 | Amazon_Sales | 96 | 1.0382 | 0.7975 | Cross_1 | 96 | 0.7275 | 0.7296 |
| Indian_Climate_Humidity | 24 | 1.0232 | 0.8742 | Coffee_Sales | 24 | 3.4196 | 1.804 | Cross_2 | 24 | 0.6832 | 0.7568 |
| Indian_Climate_Humidity | 48 | 1.1192 | 0.9234 | Coffee_Sales | 48 | 3.347 | 1.7754 | Cross_2 | 48 | 0.4517 | 0.538 |
| Indian_Climate_Humidity | 96 | 1.1471 | 0.9181 | Coffee_Sales | 96 | 3.156 | 1.6173 | Cross_2 | 96 | 0.9623 | 0.8086 |
| Indian_Climate_Temperature | 24 | 1.0286 | 0.918 | Dominic_Dataset | 24 | 0.3751 | 0.4401 | Cross_3 | 24 | 0.2549 | 0.3828 |
| Indian_Climate_Temperature | 48 | 1.0614 | 0.8827 | Dominic_Dataset | 48 | 0.4657 | 0.3777 | Cross_3 | 48 | 0.2026 | 0.3503 |
| Indian_Climate_Temperature | 96 | 1.0237 | 0.8739 | Dominic_Dataset | 96 | 1.0606 | 0.4575 | Cross_3 | 96 | 0.5052 | 0.5584 |
| Indian_Climate_Wind | 24 | 1.2106 | 0.9383 | Online_Retail | 24 | 5.1241 | 1.6251 | Cross_4 | 24 | 0.3183 | 0.4666 |
| Indian_Climate_Wind | 48 | 1.3136 | 0.9883 | Online_Retail | 48 | 3.2703 | 1.2814 | Cross_4 | 48 | 0.3274 | 0.4901 |
| Indian_Climate_Wind | 96 | 1.1546 | 0.9375 | Online_Retail | 96 | 2.0988 | 0.9942 | Cross_4 | 96 | 0.9552 | 0.8039 |
| Jena_Climate_H20C | 24 | 0.7598 | 0.8432 | Product_1 | 24 | 0.6292 | 0.6473 | Cross_5 | 24 | 0.1685 | 0.3206 |
| Jena_Climate_H20C | 48 | 1.1281 | 1.0282 | Product_1 | 48 | 0.6915 | 0.6801 | Cross_5 | 48 | 0.3388 | 0.4645 |
| Jena_Climate_H20C | 96 | 1.43 | 1.0651 | Product_1 | 96 | 0.7058 | 0.6807 | Cross_5 | 96 | 0.7979 | 0.6908 |
| Jena_Climate_RH | 24 | 0.259 | 0.5041 | Product_2 | 24 | 1.3392 | 0.9013 | Cross_6 | 24 | 0.2433 | 0.381 |
| Jena_Climate_RH | 48 | 0.2213 | 0.4602 | Product_2 | 48 | 1.0277 | 0.8445 | Cross_6 | 48 | 0.1441 | 0.2869 |
| Jena_Climate_RH | 96 | 0.2801 | 0.5126 | Product_2 | 96 | 0.8973 | 0.7799 | Cross_6 | 96 | 0.283 | 0.4436 |
| Ozone | 24 | 0.1639 | 0.3244 | Product_3 | 24 | 0.9471 | 0.8559 | Pedestrian_Counts | 24 | 0.0271 | 0.142 |
| Ozone | 48 | 0.3733 | 0.5055 | Product_3 | 48 | 0.9303 | 0.8403 | Pedestrian_Counts | 48 | 0.0213 | 0.126 |
| Ozone | 96 | 0.2742 | 0.4232 | Product_3 | 96 | 0.8487 | 0.7755 | Pedestrian_Counts | 96 | 0.0278 | 0.1424 |
| Pollution | 24 | 0.9063 | 0.9273 | Rossmann | 24 | 0.8884 | 0.741 | Rideshare | 24 | 0.7778 | 0.869 |
| Pollution | 48 | 0.6886 | 0.7472 | Rossmann | 48 | 1.0714 | 0.7926 | Rideshare | 48 | 0.8692 | 0.9048 |
| Pollution | 96 | 1.7419 | 0.9876 | Rossmann | 96 | 1.2495 | 0.9068 | Rideshare | 96 | 0.9849 | 0.9672 |
| Saugeen_Day | 24 | 1.5539 | 0.7213 | Store_1_Item_1 | 24 | 1.2051 | 0.9026 | Traffic_Hourly | 24 | 0.1943 | 0.3716 |
| Saugeen_Day | 48 | 1.1927 | 0.6815 | Store_1_Item_1 | 48 | 1.2329 | 0.9632 | Traffic_Hourly | 48 | 0.2139 | 0.3713 |
| Saugeen_Day | 96 | 0.625 | 0.4695 | Store_1_Item_1 | 96 | 0.9288 | 0.7909 | Traffic_Hourly | 96 | 0.2495 | 0.401 |
| Sunspot | 24 | 0.0021 | 0.044 | Store_1_Item_2 | 24 | 1.0651 | 0.9435 | Traffic_Weekly | 24 | 2.1159 | 1.2938 |
| Sunspot | 48 | 0.0071 | 0.0584 | Store_1_Item_2 | 48 | 0.9592 | 0.8504 | Traffic_Weekly | 48 | 1.2372 | 0.9211 |
| Sunspot | 96 | 0.006 | 0.0619 | Store_1_Item_2 | 96 | 0.7729 | 0.7253 | Traffic_Weekly | 96 | 1.7488 | 1.0553 |
| Temp_Hourly | 24 | 1.2862 | 0.9748 | Store_1_Item_3 | 24 | 1.356 | 0.9918 | Vehicle_Trips | 24 | 1.1097 | 0.9459 |
| Temp_Hourly | 48 | 1.2445 | 0.9948 | Store_1_Item_3 | 48 | 1.049 | 0.825 | Vehicle_Trips | 48 | 1.1148 | 0.9414 |
| Temp_Hourly | 96 | 1.141 | 0.9428 | Store_1_Item_3 | 96 | 0.7756 | 0.696 | Vehicle_Trips | 96 | 1.051 | 0.927 |
| Weather | 24 | 0.1213 | 0.2762 | Store_1_Item_4 | 24 | 1.2675 | 1.0148 | | | | |
| Weather | 48 | 0.0896 | 0.235 | Store_1_Item_4 | 48 | 1.0296 | 0.8873 | | | | |
| Weather | 96 | 0.1106 | 0.2668 | Store_1_Item_4 | 96 | 0.8538 | 0.7514 | | | | |
| | | | | Store_1_Item_5 | 24 | 1.5992 | 1.0624 | | | | |
| | | | | Store_1_Item_5 | 48 | 1.2378 | 0.9033 | | | | |
| | | | | Store_1_Item_5 | 96 | 1.0093 | 0.7937 | | | | |

## B.3. Aggregated Performance and Ranking

Here we report the same ranking analyses as in Section 4.2, but using the MSE (Mean Squared Error) metric. Figure 5 shows model rankings by domain (sorted by global mean performance), and Figure 6 presents the domain-specific performance footprint under MSE. The findings mirror those observed with MAE, with only minor deviations.

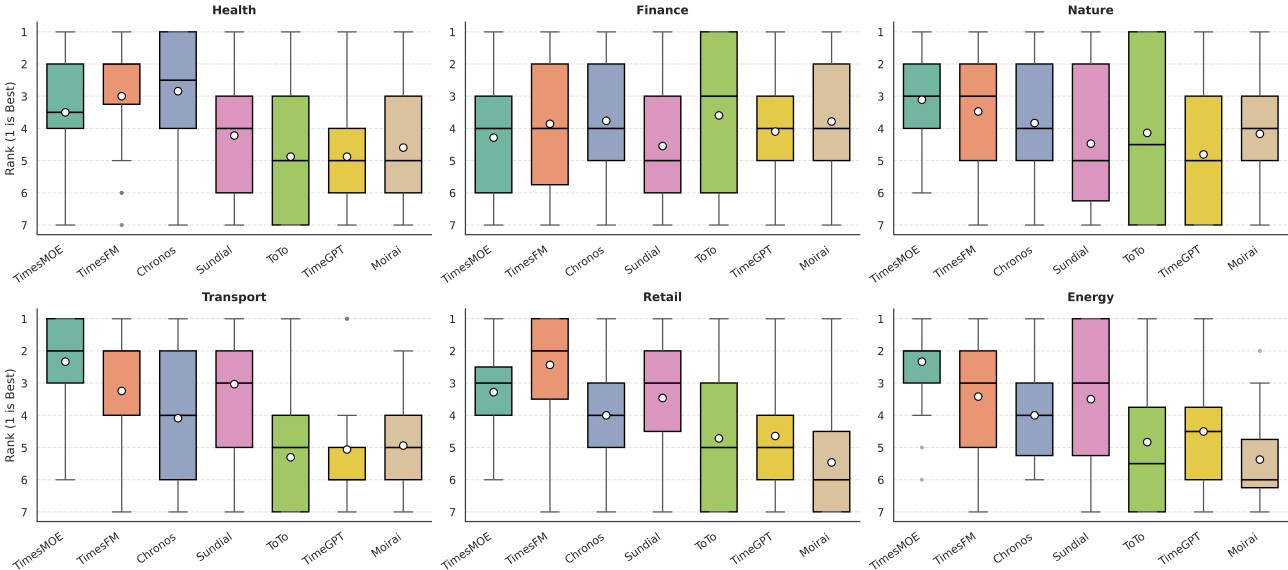

*Figure 5.* **Model Rank Distribution (MSE) Across Domains.** Box plots show the distribution of model rankings within each domain, where rank 1 indicates best performance. White circles denote mean rank; horizontal lines indicate median rank.

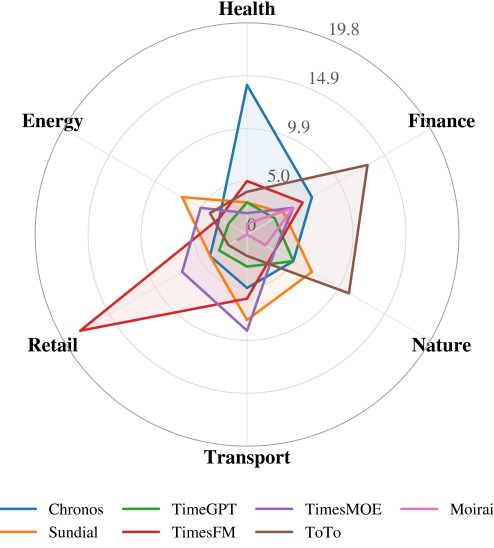

*Figure 6.* Domain-Specific Performance Footprint (MSE) of TSFMs

## C. Statistical Test for Domain-Specific Performance Differences

### C.1. Mathematical Formulation

Let $\mathcal{D} = \{d_1, \ldots, d_K\}$ denote the set of domains, and let $\mathcal{M}$ denote the set of models. For a fixed forecasting horizon $h = 24$ and error metric $E \in \{\text{MAE}, \text{MSE}\}$, we collect per-dataset errors produced by model $m \in \mathcal{M}$ within each domain $d \in \mathcal{D}$. Concretely, if domain $d$ contains $n_d$ datasets (or benchmark instances), the observed error samples are

$$\mathbf{x}_d^{(m,E)} = \{x_{d,1}^{(m,E)}, x_{d,2}^{(m,E)}, \ldots, x_{d,n_d}^{(m,E)}\}. \tag{7}$$

We test whether the distribution of errors is invariant across domains:

$$H_0 : \mathbf{x}_{d_1}^{(m,E)} \overset{d}{=} \mathbf{x}_{d_2}^{(m,E)} \overset{d}{=} \cdots \overset{d}{=} \mathbf{x}_{d_K}^{(m,E)}, \qquad H_1 : \exists\, d_i \neq d_j \text{ s.t. } \mathbf{x}_{d_i}^{(m,E)} \overset{d}{\neq} \mathbf{x}_{d_j}^{(m,E)}. \tag{8}$$

Since error distributions can be non-Gaussian and heteroskedastic across domains, we adopt the non-parametric Kruskal–Wallis test (Kruskal & Wallis, 1952). Let $N = \sum_{d \in \mathcal{D}} n_d$ be the total number of observations, and let $R(x)$ be the rank of observation $x$ among all pooled samples $\cup_d \mathbf{x}_d^{(m,E)}$ (ties are assigned average ranks). Define the mean rank of each domain as

$$\bar{R}_d = \frac{1}{n_d} \sum_{i=1}^{n_d} R\left(x_{d,i}^{(m,E)}\right). \tag{9}$$

The Kruskal–Wallis statistic is then computed as

$$H = \frac{12}{N(N+1)} \sum_{d \in \mathcal{D}} n_d \bar{R}_d^2 \;-\; 3(N+1), \tag{10}$$

which is approximately $\chi^2$-distributed with $K - 1$ degrees of freedom under $H_0$. We report the corresponding $p$-value for each model and metric, and annotate significance using the convention:

$$\text{ns} : p > 0.05, \quad {}^* : p \leq 0.05, \quad {}^{**} : p \leq 0.025, \quad {}^{***} : p \leq 0.01, \quad {}^{****} : p \leq 10^{-3}. \tag{11}$$

### C.2. Domain-Specific Performance Differences Across Horizons

#### C.2.1. HORIZON = 48

Table 14 demonstrates that domain-dependent performance differences remain statistically detectable at horizon $h = 48$ for most time-series foundation models. Under MAE, five out of seven models show significant cross-domain variation, with TIMESMOE exhibiting the strongest evidence ($p \leq 0.01$), while TIMESFM and TOTO do not reach significance. Under MSE, the domain effect is more consistent and stronger: all models achieve significance, and multiple models reach $p \leq 0.01$ (notably TIMEGPT and TIMESMOE), indicating that cross-domain shifts increasingly manifest as larger-magnitude errors at longer horizons. Overall, the results reinforce that aggregate benchmark reporting can mask domain-specific failure modes, motivating domain-aware evaluation when comparing TSFMs in realistic deployment settings.

*Table 14.* Global domain effect at Horizon = 48 using the Kruskal–Wallis test. Reported values are $p$-values with significance markers: *ns* ($p > 0.05$), $^*$ ($p \leq 0.05$), $^{**}$ ($p \leq 0.025$), $^{***}$ ($p \leq 0.01$).

| Model | MAE ($p$, sig.) | MSE ($p$, sig.) |
|---|---|---|
| Chronos | 0.0402* | 0.0131** |
| Moirai | 0.0492* | 0.0106** |
| Sundial | 0.0216** | 0.0125** |
| TimeGPT | 0.0294* | 0.0064*** |
| TimesFM | 0.0806 (*ns*) | 0.0282* |
| TimesMOE | 0.0056*** | 0.0047*** |
| ToTo | 0.0528 (*ns*) | 0.0248** |

C.2.2. HORIZON = 96

Table 15 shows that domain-dependent performance differences become less uniformly detectable at the long-horizon setting ($h = 96$). Under MAE, only MOIRAI and TIMESMOE exhibit statistically significant cross-domain variation, whereas CHRONOS, SUNDIAL, TIMEGPT, TIMESFM, and TOTO are not significant. Under MSE, evidence of domain dependence persists for SUNDIAL, TIMEGPT, MOIRAI, and TIMESMOE, while CHRONOS, TIMESFM, and TOTO remain non-significant. Overall, the weakening of significance at $h = 96$ suggests that long-range uncertainty may dominate forecasting errors for some models, effectively compressing domain-specific separability; nevertheless, the consistent significance of TIMESMOE indicates that certain architectures still retain strong domain sensitivity even at extended horizons.

*Table 15.* Global domain effect at Horizon = 96 using the Kruskal–Wallis test. Reported values are $p$-values with significance markers: *ns* ($p > 0.05$), * ($p \leq 0.05$), ** ($p \leq 0.025$), *** ($p \leq 0.01$).

| Model | MAE ($p$, sig.) | MSE ($p$, sig.) |
|---|---|---|
| Chronos | 0.2064 (*ns*) | 0.0668 (*ns*) |
| Moirai | 0.0242** | 0.0184** |
| Sundial | 0.0544 (*ns*) | 0.0293* |
| TimeGPT | 0.0518 (*ns*) | 0.0213** |
| TimesFM | 0.5652 (*ns*) | 0.4340 (*ns*) |
| TimesMOE | 0.0077*** | 0.0077*** |
| ToTo | 0.4542 (*ns*) | 0.2521 (*ns*) |

C.2.3. TRENDS IN DOMAIN DEPENDENCE IN DIFFERENT HORIZONS

Across horizons, the Kruskal–Wallis tests reveal a clear attenuation of domain-dependent performance differences as the forecast horizon increases from $h = 24$ to $h = 96$.

At short horizon ($h = 24$), domain effects are pervasive: nearly all models reject the null hypothesis under MSE, and most remain significant under MAE, indicating that cross-domain statistical heterogeneity (e.g., noise structure, seasonality strength, and missingness patterns) translates into measurable differences in error distributions. At mid-range horizon ($h = 48$), the signal remains strong under MSE (all models significant), while MAE begins to weaken for some methods (e.g., TIMESFM, TOTO), suggesting that domain shifts increasingly manifest through occasional large deviations rather than systematic median-level shifts.

By long horizon ($h = 96$), significance becomes sparse and model-dependent: several methods become non-significant for both metrics, consistent with the hypothesis that long-range uncertainty and compounding multi-step errors increasingly dominate the loss, inflating error variance and causing domain-specific effects to be masked by a higher noise floor. Consequently, rank separation across domains becomes weaker at larger horizons, yielding systematically reduced statistical confidence (larger $p$-values) as $h$ increases.

Overall, the horizon-wise trend indicates that domain-aware evaluation is most critical at short-to-mid horizons where cross-domain differences are strongest, while at extended horizons only a subset of models continues to exhibit statistically distinguishable domain-specific performance.

