# OpenReview forum: "Position: Time-Series Foundation Models Require Explicit Domain-Level Benchmarks"
_ICML.cc/2026/Position_Paper_Track — ICML 2026 Position Paper Track regular_

### Official Review · Reviewer_GFYB · 2026-02-14

**Significance:** 2
**Argument Clarity:** 3
**Rating:** 5
**Confidence:** 4

**Questions:**

See my weaknesses.

**Alternative Views Section:**

Yes

**Compliance With Llm Reviewing Policy A Conservative:**

Affirmed.

**Discussion Potential:**

2

**Final Justification:**

Thanks for the authors for taking my questions seriously. My questions are fully solved.

Since my rating was already positive, I'll keep my score.

**Paper Summary:**

The current time series benchmarks pool datasets from many domains with uneven representation. The paper argues that the inherent variance in different domains of time series data create challenges for universal models and benchmarking. The experimental results also demonstrate existing times series foundation models (TSFMs) demonstrate different domian wise performance. As a results, the position propose that the TSFM require explicit domain-specific benchmarks to facilitate evaluation of models in different application area.

**Position:**

Yes

**Position In Title:**

Yes

**Related Work:**

3

**Strengths And Weaknesses:**

Strengths:
1. The writing is clear and easy to follow.
2. The paper clearly demonstrated a limitation to the current benchmarking, making the position thoughtful and timely contribution to the community.
3. The experimental results are extensive and rigorous.

Weaknesses:
I don't find any significant weaknesses. Instead, I arise two questions to the authors, and I believe solving these questions would improve the paper.
1. The granularity of the benchmarking needs further consideration. The current position suggests dividing benchmarking into domain-specific evaluations, and the results show that the distribution of model “wins” shifts across domains. However, if we further split the evaluation into finer granularity (e.g., per dataset), would the distribution of model wins also differ across datasets? If so, is the variability in performance truly driven by domain knowledge, or simply by dataset variation, or both? Also, based on Figure 3, it is hard to conclude that a model “wins a domain” given the high variance. To my understanding, the performance differences are not statistically significant to conclude for Figure 4.
2. Does the domain-specific knowledge/structure affect the evaluation itself? The current position focuses on datasets, but I encourage the authors to consider: instead of evaluating models uniformly across domains, should the evaluation metrics also vary by domain?

**Support:**

3

---

> ### Author Rebuttal · Authors · 2026-03-29
>
> Thank you for your thoughtful and constructive review. We appreciate your recognition of the clarity of our writing, the relevance of the problem, and the importance of improving time series benchmarking. We appreciate your recognition of our thorough testing across various models and domains.
>
> ---
>
> Below, we address each of your questions in detail:
>
> **Q1.  Granularity of Benchmarking and Statistical Significance**
>   - *Response:* You raise an important point about the source of variance. We do not claim a single model dominates a domain; instead, performance is fragmented and domain-dependent. To test whether this reflects domain effects rather than random dataset variation, we conducted formal statistical analysis, which confirms domain-level significance:
>       1.  *Statistical Proof of Domain Effect:* We utilized the non-parametric Kruskal-Wallis test (**Table 3** in the main text, and **Tables 10 & 11** in Appendix C) to test the null hypothesis that error distributions are invariant across domains. For both MAE and MSE, most models yield statistically significant evidence ($p \le 0.01$ for several models under MSE at horizon 24) rejecting the null hypothesis. This provides quantitative proof that the error distributions vary systematically *across application domains* and are not merely the result of random per-dataset noise.
>       2.  *The High Variance in Figure 3:* We agree that Figure 3 shows high variance. In fact, this high variance is central to our argument. It illustrates that even within a specific domain, inductive biases behave inconsistently. If performance is this volatile *within* a defined domain, aggregating these scores into a single global benchmark average (as is currently standard practice) is mathematically misleading and actively hides this volatility from practitioners.
>       3.  *The Diagnostic Value of Intra-Domain Variance:* Even when variance exists within a domain, domain stratification remains useful because it localizes and exposes this variability instead of hiding it in global averages. In a pooled benchmark, strong performance on large or easy datasets can mask failures on harder cases, making reliability in any specific application unclear. By isolating domains, the variance becomes interpretable and actionable, enabling users to assess model reliability within their target setting and motivating finer-grained analyses based on structural characteristics.
>
>       Overall, this ultimately supports our central position that domain-level evaluation is necessary to reveal and interpret systematic variability that pooled benchmarks obscure. We will include these discussions in the final paper.
>
>
>
> **Q2.  Domain-Specific Evaluation Metrics**
>   - *Response:* This is an insightful observation, and we agree that domain-specific benchmarks should influence evaluation metrics. Different domains prioritize different goals. For example, finance often focuses on directional accuracy or risk-adjusted returns, while healthcare emphasizes calibration and timely detection under missingness. In this work, we use standard metrics such as MSE and MAE to maintain consistent and comparable evaluation across benchmarks and to isolate domain effects. We will add a call to action in Section 7 to highlight the need for future benchmarks to include domain-aligned metrics that better reflect real-world use.
>       - *Proposed **Section 7. A4. Domain-Aligned Evaluation Metrics.*** Stratifying datasets by domain is only the first step. To measure real-world utility, benchmarks must move beyond uniform error metrics (e.g., MSE, MAE) and adopt domain-aligned criteria reflecting operational needs. For example, finance often prioritizes directional accuracy or Sharpe-ratio–based measures, healthcare emphasizes AUROC for clinical decision tasks and calibration under missingness, and energy forecasting commonly uses CRPS for probabilistic load prediction. Future benchmarks should pair domain-specific datasets with metrics used in practice. Prior work such as MIRA [1] and Fincast [2] further shows that domain-specific inductive biases in architecture significantly improve performance in healthcare and finance, reinforcing the need for domain-aware evaluation design.
>
>
> ---
>
> ### References
>
> 1.  Li et al., MIRA: Medical time series foundation model for realworld health data. NeurIPS 2025
> 2. Zhu et al. Fincast: A foundation model for financial time-series forecasting. CIKM 2025
>
> ---
>
> We sincerely thank the reviewer for the careful reading and constructive feedback, which helped us further clarify key aspects of our position and evaluation framework. We hope our responses have adequately addressed the concerns and strengthened the clarity and motivation of our work. We would be grateful if the reviewer could consider updating the ratings in light of these clarifications.

---

> > ### Author Rebuttal · Reviewer_GFYB · 2026-04-01
> >
> > Thanks for the authors for taking my questions seriously. My questions are fully solved.
> >
> > I also think the author did a good job responding to Reviewer `EU1r`. Good luck.

---

### Official Review · Reviewer_f45w · 2026-03-11

**Significance:** 3
**Argument Clarity:** 4
**Rating:** 5
**Confidence:** 4

**Questions:**

1. Chronos actually performs quite well cross different domains in the experiments. In particular, the recent Chronos-2 further improves the forecasting performance significantly in comparison to Chronos. Why do you not try Chonos-2?

2. An underlying assumption is that tasks in the same domain share some similarities, so the domain is a natural “class label” for categorize different forecasting tasks. Personally, I accept the use of domain. But in some domains, the data and forecasting task may exhibit quite diverse characteristics, while some other domains may be simpler. In this case, a more detailed treatment beyond “domain” may be preferred.

**Alternative Views Section:**

Yes

**Compliance With Llm Reviewing Policy A Conservative:**

Affirmed.

**Discussion Potential:**

3

**Paper Summary:**

As time series foundation model (TSFM) are more popular and achieves expressive performance on major benchmarks, this paper argues that time series foundation models require domain-level benchmark and evaluation, although they have achieved strong performance on aggregated benchmarks. In terms of empirical studies, it evaluates seven existing time series foundation models across 72 datasets from 6 domains, including healthcare, finance, energy, nature, transport, and retail. The experimental results (Tables 2, 5~9) show that: 1) no single model can dominate across all domains; 2) model rankings vary widely across different domains; 3) statistical tests (Kruska-Wallis) confirms that performance difference cross different domains are significant.

It provides insightful analysis on domain-specific challenges for time series foundation models, including inherent variance in time series data, temporal structure and sampling irregularity, stationary, concept drift, and distribution shifts. Indeed, the effective hypothesis class for forecasting varies across domains.

In terms of actions, the paper calls for domain-stratified evaluation, evaluating domain-aware models, and cross-domain transfer framework, all focusing on TSFM for different domains.

Overall, the authors’ perspective on time series foundation model seems pessimistic, but the point that the evaluation on aggregated benchmarks should be improved to domain-specific benchmark and evaluation is reasonable and can further guide the development of time series foundation model.

**Position:**

Yes

**Position In Title:**

Yes

**Related Work:**

3

**Strengths And Weaknesses:**

Strengths:
1. It addresses an important and fundamental problem for time series foundation model. I strongly agree that aggregated metrics only may be misleading in developing time series forecasting models.
2. Solid experiments for popular TSFM models and benchmarks across different domains support the domain-specific difference for different TSFM models.
3. Generally good theoretical analysis.

Weaknesses or suggestions:
1. It lacks the evaluation of more recent time series foundation model, such as Chonos-2 and TabPFN-TS.
2. Limited analysis why different TSFM models achieve different perofmrance in different domains.

**Support:**

3

---

> ### Author Rebuttal · Authors · 2026-03-30
>
> Thank you very much for your thoughtful review. We sincerely appreciate your recognition of the importance of addressing limitations in aggregated evaluation for time series foundation models and your agreement that this is a meaningful direction for the field. We are also grateful for your acknowledgment of the strength of our experimental setup and results across multiple models and datasets.
>
> ---
>
> Below, we address each of your concerns and questions in detail:
>
> - **W1/Q1.  Inclusion of more recent time series models (Chonos-2 and TabPFN-TS)**
>   - *Response:* Thank you for the concern. We indeed evaluated Chronos-2 in our experiments. In the main tables, it is labeled as Chronos, while Appendix A.2.3 (Table 4) specifies the exact variants used for all models, including Chronos-2. We acknowledge this ambiguity and will update all mentions to explicitly state Chronos-2 in the revised manuscript to ensure clarity and consistency.
>   - Regarding *TabPFN-TS*, we agree it is a highly relevant recent model and the results are attached below for energy domain for `horizon = 24` because of space contraint. Comprehensive results are available here:  https://freeimage.host/i/B2ATYOv
>
>
> | Dataset | Horizon | MSE | MAE |
> |---|---|---|---|
> | AUS Elec Demand | 24 | 0.0437 | 0.1796 |
> | Electricity Weekly | 24 | 0.5413 | 0.3332 |
> | ETTh1 | 24 | 0.1669 | 0.3287 |
> | ETTh2 | 24 | 0.1813 | 0.3761 |
> | ETTm1 | 24 | 0.2187 | 0.3526 |
> | ETTm2 | 24 | 0.0739 | 0.2296 |
> | London SmartMeters | 24 | 1.9348 | 0.9312 |
> | Solar 10min | 24 | 0.0013 | 0.0236 |
>
>
>
> Across six domains, TabPFN-TS demonstrates clear strengths in Health and Energy, securing the most first-place results in Health on clinical, hospital, and demographic datasets, and matching or outperforming models like Sundial and TimesMOE on key Energy benchmarks. In Transport, it consistently ranks among the top 2–3 models, while Nature performance is competitive but reveals relative weakness on periodic climate signals. Finance and Retail represent the most challenging domains, where stochastic, low-signal series favor models like ToTo and Chronos, suggesting that incorporating periodicity-aware and volatility-sensitive inductive biases could address these remaining gaps in future work.
>
>
>
> - **W2/Q2.  Limited analysis why different TSFM models achieve different perofmrance in different domains**
>   - *Response:* We appreciate the reviewer’s insightful observation and agree that domain serves as a proxy for deeper statistical structure rather than a fundamental descriptor. We address the reasoning below:
>     1. *Conflicting inductive biases and optimization effects:* As discussed in Section 2.2, TSFMs encode architectural priors such as fixed-length patching that align well with regularly sampled data (e.g., energy or weather) but degrade under irregular sampling and missingness (e.g., healthcare). In addition, Section 2.1 shows that pooled pretraining induces gradient conflicts across heterogeneous data-generating processes, leading models to converge to compromises that favor dominant or smoother domains over high-variance ones.
>     2. *Beyond semantic domains to structural properties:* We agree that intra-domain heterogeneity exists and that semantic labels are an approximation of deeper structure. In this work, we use domain labels as a practical first step reflecting current evaluation practice, while our broader position argues for moving toward evaluation based on intrinsic properties such as sampling frequency, stationarity, and spectral characteristics, which better explain transfer behavior.
>     3. *Model-specific design and training differences:* We note that part of the variation also arises from differences in architecture and training data across models, as briefly discussed in the introduction. Specialized models such as MIRA [1], targeting medical irregularity and missingness, and FinCast [2], addressing financial non-stationarity, report strong gains in the specific datasets,  but are not evaluated under specialized benchmarks. The specialized benchmarks will help to understand the gain from architectural and training differences.
>
>     Overall, these points directly support our central position that aggregated benchmarks obscure fundamentally different learning regimes, and that explicit domain-level (and ultimately structure-level) evaluation is necessary for reliable assessment of TSFMs. We will include these discussions in the final paper.
>
> ---
>
> ### References
>
> 1.  Li et al., MIRA: Medical time series foundation model for realworld health data. NeurIPS 2025
> 2. Zhu et al. Fincast: A foundation model for financial time-series forecasting. CIKM 2025
>
>
> ---
>
>
> We sincerely thank the reviewer for the careful reading and constructive feedback, which helped clarify important aspects of our work. We hope our responses address the concerns and strengthen confidence in the validity, relevance, and significance of our findings.

---

> > ### Author Rebuttal · Reviewer_f45w · 2026-04-04
> >
> > 1. Thanks for the clarification of Chronos-2.
> > 2. The new experiments of TabPFN-TS sounds good. It is consistent with what I observed in my practice: generally it is a very competitive foundation model, but underperforms other foundation models in retail and finance. It is strongly recommended to incorporate experimental results of TabPFN-TS in the revision.
> > 3. Thanks for providing the analysis explaining different performance in different domains. Generally I agree with the analysis in the rebuttal. Given the short rebuttal time, I can understand the difficulty to explore it throughly. I would recommend considering it as the futher work.

---

### Official Review · Reviewer_rH4e · 2026-03-12

**Significance:** 3
**Argument Clarity:** 3
**Rating:** 4
**Confidence:** 3

**Questions:**

See Weakness

**Alternative Views Section:**

Yes

**Compliance With Llm Reviewing Policy A Conservative:**

Affirmed.

**Discussion Potential:**

3

**Paper Summary:**

This paper puts forward the position that time series foundation models require precise domain-level benchmarks. Existing benchmark datasets for time series foundation models typically cover multiple domains and involve heterogeneous representations, which may obscure the true performance of these models. The paper outlines several challenges in building domain-general models and highlights a serious domain imbalance problem in current time series foundation model benchmarks, which limits accurate model evaluation. Finally, the authors conduct experiments on datasets partitioned by domain and find that models leading the overall leaderboard do not necessarily perform well within specific domains. This observation suggests that domain-level evaluation methods are necessary.

**Position:**

Yes

**Position In Title:**

Yes

**Related Work:**

3

**Strengths And Weaknesses:**

S1: The paper is logically rigorous, well written, and the transitions between paragraphs are natural. It effectively explains the limitations of existing benchmarks for time series foundation models and the challenges involved in building universal foundation models.

S2: The experiments are sufficient, and the conclusions effectively support the paper’s main claims.

W:
Existing benchmarks are designed to evaluate the general predictive capability of time series foundation models; therefore, they include datasets from multiple domains. However, some domains (e.g., healthcare) contain relatively few datasets, possibly due to the difficulty of data collection. The paper proposes constructing domain-level benchmarks to help determine whether time series foundation models can be applied to specific domains. However, when the amount of available data in a domain is very limited, it remains unclear whether such benchmarks can effectively evaluate the practical value and true capability of the models.

**Support:**

3

---

> ### Author Rebuttal · Authors · 2026-03-29
>
> Thank you very much for your thoughtful review. We sincerely appreciate your recognition of the paper’s clarity, logical structure, and effective explanation of the limitations of existing time series foundation model benchmarks and domain imbalance issues. We are also grateful for your acknowledgment that our experimental results are sufficient and support the main claims of the paper.
>
> ---
>
> Below, we address your concern in detail:
>
> **W1: Data Scarcity and Domain-Specific Benchmark Development:**
>
> We appreciate the reviewer highlighting this critical point. You are absolutely correct that certain domains, particularly healthcare, suffer from severe data scarcity due to collection difficulties and privacy constraints. However, rather than diminishing the value of domain-level benchmarks, we argue that this scarcity is precisely what makes explicit domain-stratification mathematically and practically essential.
>
> Below, we address why stratified evaluation is the only way to measure true capability in limited-data domains, drawing directly from the evidence in our manuscript:
>
> 1. *Data Scarcity Exacerbates the "Masking" Problem.*
> As detailed in **Table 1** of our manuscript, healthcare represents merely 0.7% of the series and 0.1% of the total observations in GIFT-Eval. Conversely, Traffic/Transport and Web data heavily dominate these benchmarks (e.g., contributing 45.2% and 41.9% of TSFM-Bench, respectively). When benchmarks are pooled globally, the performance on a data-scarce domain like healthcare is mathematically swallowed by the massive volume of traffic or web data. A model can fail catastrophically on clinical forecasting yet still rank as the #1 model globally. Therefore, separating these scarce domains is the only way to expose whether a model possesses any practical value in that field.
>
> 2. *True Capability is Structural, Not Just Volumetric*.
> The practical value of a model in a specific domain is not strictly tied to the benchmark's size, but to the benchmark's ability to test domain-specific structural challenges. As outlined in **Section 2.2**, healthcare data is uniquely defined by multi-rate sampling and informative missingness (e.g., irregular lab tests vs. continuous ECGs). A targeted domain-level benchmark, even if small in absolute volume, effectively evaluates whether a TSFM's inductive biases (like fixed-length patch tokenization) break down under these irregular conditions. Our Kruskal-Wallis statistical tests (**Section 4.3**) prove that even with limited datasets, the error distributions are statistically separable and domain-dependent, proving the evaluation carries significant signal.
>
> 3. *Enabling the Evaluation of Domain-Aware Specialists.*
> As we advocate in our Call to Action (**Section 7, A2**), establishing these benchmarks is crucial for the research community to compare general TSFMs against emerging domain-specialized models. For instance, without an explicit (albeit small) healthcare benchmark, it is impossible to rigorously quantify if a specialized architecture like MIRA [1], which is designed specifically for medical irregularity, genuinely outperforms a massively scaled universal model like TimesFM or Chronos in that constrained environment.
>
> Overall, we believe that data scarcity in certain domains does not weaken the case for domain-level benchmarks but instead makes them more necessary for revealing masked failures and true domain-specific capability. We further expect that our study will encourage domain-specific evaluation and provide incentives for researchers to release more preprocessed datasets as benchmarks in their respective domains for TSFM research, especially as more research labs publish their own models. We hope this clarification addresses the reviewer’s concern and strengthens our position on the importance of stratified evaluation. We will include these discussions in the final paper.
>
> ---
>
> ### References
> 1.  Li et al., MIRA: Medical time series foundation model for realworld health data. NeurIPS 2025
>
> ---
>
> We sincerely thank the reviewer for the careful reading and insightful feedback, which helped us clarify key aspects of our work. We hope our responses have adequately addressed the concerns and provided further clarity on our position and experimental findings. We would be very grateful if the reviewer could consider improving the overall rating and updating the confidence score in light of our clarifications.

---

> > ### Author Rebuttal · Reviewer_rH4e · 2026-04-04
> >
> > Thanks for the response, my concerns are addressed.

---

### Official Review · Reviewer_EU1r · 2026-03-13

**Significance:** 2
**Argument Clarity:** 2
**Rating:** 3
**Confidence:** 5

**Questions:**

1. Section 2 raises several theoretical challenges (e.g., irregular sampling, spectral differences, concept drift), but these are not analyzed in the experiments. Could the authors clarify how the empirical results relate to these challenges?
2. Could the authors provide more detailed descriptions of the datasets used (e.g., sample sizes, sequence counts, and key characteristics)?
3. Have the authors considered including additional TSFM baselines such as MOMENT or Sundial in the evaluation? If not, could they comment on how these models might behave under the proposed evaluation protocol?
4. The challenges outlined in Section 2 (e.g., irregular sampling, spectral differences, concept drift) are not analyzed in the experiments. Why?

**Alternative Views Section:**

Yes

**Compliance With Llm Reviewing Policy A Conservative:**

Affirmed.

**Discussion Potential:**

2

**Final Justification:**

This is a comprehensive dataset and benchmark paper. The authors have addressed my concerns. I have raised my score from 2 to 3. The reason for weak reject is because I am not sure about the suitability of the paper for this track. I shall leave it to AC/SAC to decide. Overall, it serves as a comprehensive evaluation / benchmark paper.

**Paper Summary:**

This paper examines the evaluation of Time Series Foundation Models (TSFMs), pointing out that current mainstream time series benchmarks typically adopt cross dataset pooled evaluation, which may obscure genuine differences in model capabilities due to imbalanced data distributions across different application domains. To address this, the authors collect 72 time series datasets spanning 6 application domains and evaluate 7 TSFM models. By comparing pooled evaluation with domain level evaluation, they find significant variations in model performance and rankings across different domains. Based on these findings, the paper argues that relying solely on pooled benchmarks may lead to misleading model comparisons, and recommends that future TSFM benchmarks adopt domain aware evaluation reporting performance by domain and analyzing cross domain robustness.

**Position:**

Yes

**Position In Title:**

Yes

**Related Work:**

2

**Strengths And Weaknesses:**

Strengths:
1. This work identifies domain imbalance and aggregation bias in time series foundation model evaluation. As TSFMs continue to develop, designing fair and trustworthy benchmarks is a critical issue, making this work practically relevant.
2. The paper conducts systematic evaluations using 72 datasets, 6 domains, and 7 models, offering broad coverage. This scale of experimentation helps comprehensively observe how models perform across different application areas.
3. Beyond reporting average performance, the paper analyzes:
- Domain level rankings
- Cross domain performance disparities
- Statistical significance testing
These analyses help reveal important phenomena that benchmark aggregation might conceal.
4. The recommendations for the community are valuable: reporting results by domain, analyzing cross domain robustness, and avoiding single pooled leaderboards.

Weaknesses:
1. I suspect this work might not fit the position track as well as it seems that it actually reads more like a solid empirical benchmark study. My reasons are: First, the claim that existing pooled benchmarks mask domain specific failure modes and that cross domain scores and rankings are unstable is essentially empirical observation rather than a principled stance. Second, advocating for domain stratified evaluation in benchmarking feels more like a methodological recommendation for time series forecasting setup rather than a research community position. Third, Section 2 lists numerous theoretical challenges (irregular sampling, concept drift, etc.) to justify domain stratified evaluation, yet the paper ultimately returns to experimental the standard format for benchmark tracks.

In short, the paper does have a position, but it's overshadowed by a benchmark style empirical study, making its track fit less pure. The work quality and advocacy are beneficial to the community, though I'd prefer to see it moved to the benchmark track.

2. Section 2 mentions various challenges, but these aren't analyzed or interpreted in the benchmark evaluation itself. For instance, there are no experiments measuring the impact of irregular sampling, spectral structure differences, or concept drift. Gradient conflict is a training dynamics issue largely unrelated to this work. While the paper proves that domain matters, connecting these analyses to the earlier theoretical challenges would strengthen the argument.

3. This isn't a new dataset, new benchmark, or new metric and it’s an evaluation protocol change. The paper claims current benchmarks fail due to domain imbalance, but the domain dependency of model rankings is common knowledge.

4. The paper aims to address dataset imbalance, yet I couldn't find detailed dataset descriptions like the sample sizes, sequence volumes, dataset characteristics, or spatial/temporal distributions.
5. The baselines are comprehensive, but I'm still curious about results for other well known models like MOMENT[1] and Sundial[2] in this experimental setup.

[1] Goswami M, Szafer K, Choudhry A, et al. MOMENT: A Family of Open Time-series Foundation Models, Forty-first International Conference on Machine Learning.

[2] Liu Y, Qin G, Shi Z, et al. Sundial: A Family of Highly Capable Time Series Foundation Models, International Conference on Machine Learning. PMLR, 2025: 39295-39317.

**Support:**

3

---

> ### Author Rebuttal · Authors · 2026-03-30
>
> Thank you for your thoughtful and constructive review. We appreciate your recognition of the importance of addressing domain imbalance and aggregation bias in TSFMs, as well as the scale and depth of our experiments and analyses.
>
> ---
>
> Below, we address each of your concerns and questions in detail:
>
> - **W1.  Suitability of Position Paper Track:**
>   - *Response:* Thank you for your thoughtful comment. We respect your perspective and understand that rigorous empirical results are typical for benchmark tracks. Our perspective is different: we are not proposing a new benchmark but rather advocating for domain-specific benchmarks, which aligns with the ICML Position Paper track. Our use of large-scale experimentation to expose structural flaws follows prior position papers [1,2,3,4], which similarly combine empirical analysis with conceptual arguments.
>
> - **W2/Q1/Q4.  Analysis of Theoretical Challenges:**
>   - *Response:* We appreciate the reviewer's comment and the expectation for more direct experimental analysis on differences between domains. Since the distinction between domains is already well-established in prior literature, we presented these findings in detail in **Section 2**, where we review how each domain exhibits distinct time series properties. To strengthen this evidence, we conducted experiments comparing model performance across different domains and within the same domain, and applied the Kruskal-Wallis test to support our hypothesis.
> - **W3.  Contribution and Common Knowledge Concerns:**
>   - *Response:*  We appreciate the reviewer's perspective and agree that domain dependency is intuitively recognized. Yet this intuition has not spurred advocacy for domain-specific benchmarking in the time series foundation model community, where pooled benchmarks remain dominant. Furthermore, no comprehensive study has supported this intuition until ours. We see this as a much-needed position to demonstrate that domain differences are not merely assumptions but empirically grounded realities that call for domain-specific benchmarks. This is especially timely given the emergence of domain-specific models like MIRA and FINCAST, for which no proper benchmarks currently exist.
> - **W4/Q2.  Dataset Imbalance Concerns and Details:**
>   - *Response:* We sincerely apologize if the dataset details were not sufficiently clear in the current draft. Table 1 reports dataset imbalance, sample sizes, and sequence counts, showing representation skew across domains in GIFT-Eval and TSFM-Bench. Appendix A.1 describes all 72 datasets, including key properties; for example, Healthcare includes both low-frequency epidemiological series (e.g., Chickenpox) and high-frequency physiological signals (e.g., MIT-BIH T1–T4), Energy spans multiple temporal resolutions (e.g., Solar 10min vs. Electricity Weekly), and Finance covers diverse asset classes and behavioral patterns. We agree that a more explicit structural characterization would improve clarity and will expand Appendix A.1 into a unified table with sample size, seasonality, trend, and frequency. We provide energy domain properties as an illustrative example here: https://freeimage.host/i/B2Aqf1f
> - **W5/Q3.  Suggested Extra Models:**
>   - *Response:* Thank you for highlighting these models. Sundial is already included in our baseline set with results reported in the manuscript. For MOMENT, we evaluated it on the all domain, it performs well in Retail and Nature, matching top models on structured sales and climate data. It lags behind in Transport and Energy, where Sundial and TimesMOE achieve much lower MAE and MSE. Finance and Health are the hardest domains, where noisy patterns and irregular events favor models built to handle volatility and sharp shifts, areas where MOMENT still needs improvement.
>
> **Energy domain results on MOMENT (Horizon = 24).** (Full Results:  https://freeimage.host/i/B2AzXEX)
>
> |*Dataset|MSE|MAE|
> |-|-|-|
> | AUS Elec Demand| 0.2752 | 0.4476 |
> | Electricity Weekly | 1.4592 | 1.0324 |
> | ETTh1| 2.0849 | 1.3048 |
> | ETTh2| 0.6918 | 0.7373 |
> | ETTm1| 0.9809 | 0.8259 |
> | ETTm2 | 0.6748 | 0.7947 |
> | London SmartMeters| 3.7717 | 1.4318 |
> | Solar 10min| 0.2422 | 0.4794 |
>
>
> ---
>
> ### References
> 1. Alaa et al. (2025) Position: Medical Large Language Model Benchmarks Should Prioritize Construct Validity, ICML'25 Position Paper (Oral)
> 2. Bechler-Speicher et al. (2025) Position: Graph Learning Will Lose Relevance Due To Poor Benchmarks, ICML'25 Position Paper
> 3. Roy et al. (2025) Position: Graph Matching Systems Deserve Better Benchmarks, ICML'25 Position Paper
> 4. Riemer et al. (2025) Position: Theory of Mind Benchmarks are Broken for Large Language Models, ICML'25 Position Paper
>
> ---
>
> We sincerely thank the reviewer for the careful reading, constructive feedback, and valuable suggestions. We would be grateful if the reviewer could consider increasing the scores in light of our clarifications and additional analyses.

---

> > ### Author Rebuttal · Reviewer_EU1r · 2026-04-03
> >
> > Thank you for your comprehensive rebuttal and effort with the additional analysis and results. I still doubt the suitability of the paper to this track, however I'm happy to increase my score.

---

### Decision · Program_Chairs · 2026-04-30

**Decision:**

Accept (regular)

**Comment:**

The paper does a nice job of discussing many common ways that time-series settings vary by domain, including concept drift, distribution shift, variances in temporal structure, etc.  It that quantifies the differences in performance of different methods across domains.  As someone who works in this field, I found the arguments convincing and striking.  We would like to think we can develop general purpose time-series foundation models, but to know if we have succeeded at that we need domain specific evaluations across many domains, and it simply may not be possible to develop such models that are better than those fine-tuned to certain domains.  In any case, the evaluations should be domain specific so we will know.